# Heterozygous variants in *PLCG1* affect hearing, vision, cardiac, and immune function

Mengqi Ma[1,2†], Yiming Zheng[1,2†‡], Mingxi Deng[1,2], Shenzhao Lu[1,2], Xueyang Pan[1], Xi Luo[1], Michelle Etoundi[2,3], David Li-Kroeger[2,3], Kim C Worley[1], Lindsay C Burrage[1], Lauren S Blieden[4], Aimee Allworth[5§], Wei-Liang Chen[3,5#], Giuseppe Merla[6,7], Barbara Mandriani[8], Catherine E Otten[9], Pierre Blanc[10], Jill A Rosenfeld[1], Debdeep Dutta[2¶], Shinya Yamamoto[1,2], Michael F Wangler[1,2], Ian A Glass[5,11,12], Jingheng Chen[5], Elizabeth Blue[5,12,13], Paolo Prontera[14], Jeremie Rosain[15,16], Sandrine Marlin[17,18], Seema R Lalani[1], Hugo J Bellen[1,2*], Undiagnosed Diseases Network

[1]Department of Molecular and Human Genetics, Baylor College of Medicine, Houston, United States; [2]Jan and Dan Duncan Neurological Research Institute at Texas Children's Hospital, Houston, United States; [3]Department of Neurology, Baylor College of Medicine, Houston, United States; [4]The Cullen Eye Institute, Department of Ophthalmology, Baylor College of Medicine, Houston, United States; [5]Division of Medical Genetics, Department of Medicine, University of Washington School of Medicine, Seattle, United States; [6]Laboratory of Regulatory & Functional Genomics, Fondazione IRCCS Casa Sollievo della Sofferenza, San Giovanni Rotondo, Italy; [7]Department of Molecular Medicine & Medical Biotechnology, University of Naples Federico II, Naples, Italy; [8]Department of Interdisciplinary Medicine, University of Bari "Aldo Moro", Bari, Italy; [9]Department of Neurology, University of Washington and Seattle Children's Hospital, Seattle, United States; [10]SeqOIA Genomics Platform, Assistance Publique–Hôpitaux de Paris (AP-HP), Paris, France; [11]Division of Genetic Medicine, Department of Pediatrics, University of Washington School of Medicine, Seattle, United States; [12]Brotman Baty Institute, Seattle, United States; [13]Institute for Public Health Genetics, University of Washington, Seattle, United States; [14]Medical Genetics and Rare Diseases Unit, Hospital of Perugia, Perugia, Italy; [15]Laboratory of Human Genetics of Infectious Diseases, Imagine Institute, Necker Hospital for Sick Children, Paris, France; [16]Center for the Study of Immune Deficiencies, Necker-Enfants Malades Hospital, AP-HP Centre, University of Paris, Paris, France; [17]Genetics of Rare Ophthalmological, Auditory and Mitochondrial Disorders, Inserm UMR_S1163, Imagine Institute, Paris, France; [18]Reference Center for Genetic Deafness, Department of Genomic Medicine for Rare Diseases, Necker-Enfants Malades Hospital, AP-HP Centre, University of Paris, Paris, France

*For correspondence:
hbellen@bcm.edu

†These authors contributed equally to this work

Present address: ‡State Key Laboratory of Cellular Stress Biology, School of Life Sciences, Faculty of Medicine and Life Sciences, Xiamen University, Xiamen, China; §Invitae Corporation, San Francisco, United States; #Children's National Medical Center and George Washington University, Washington DC, United States; ¶Department of Biological Sciences & Bioengineering, and Mehta Family Centre for Engineering in Medicine, Indian Institute of Technology Kanpur, Uttar Pradesh, India

## eLife Assessment

This **important** study reveals how Drosophila may be used to investigate the role of missense variants in the PLCG1 phospholipase gene in human diseases. The experimental evidence is **compelling** and brings together rigorous analysis of clinical and model organism phenotypes with a structural analysis of the PLCG1 protein.

**Abstract** Phospholipase C isozymes (PLCs) hydrolyze phosphatidylinositol 4,5-bisphosphate (PIP$_2$) into inositol 1,4,5-trisphosphate (IP$_3$) and diacylglycerol (DAG), important signaling molecules involved in many cellular processes including Ca$^{2+}$ release from the endoplasmic reticulum (ER). *PLCG1* encodes the PLCγ1 isozyme that is broadly expressed. Hyperactive somatic mutations of *PLCG1* are observed in multiple cancers, but only one germline variant has been reported. Here, we describe seven individuals with heterozygous missense variants in *PLCG1* [p.(Asp1019Gly), p.(His380Arg), p.(Asp1165Gly), and p.(Leu597Phe)] who present with hearing impairment (5/7), ocular pathology (4/7), cardiac septal defects (3/6), and various immunological issues (5/7). To model these variants *in vivo*, we generated the analogous variants in the Drosophila ortholog, *small wing* (*sl*). We created a null allele *sl*$^{T2A}$ and assessed its expression pattern. *sl* is broadly expressed, including wing discs, eye discs, and a subset of neurons and glia. *sl*$^{T2A}$ mutant flies exhibit wing size reductions, ectopic wing veins, and supernumerary photoreceptors. We document that mutant flies also exhibit a reduced lifespan and age-dependent locomotor defects. Expressing wild-type *sl* in *sl*$^{T2A}$ mutant flies rescues the loss-of-function phenotypes, whereas the variants increase lethality. Ectopic expression of an established hyperactive *PLCG1* variant, p.(Asp1165His) in the wing pouch causes elevated Ca$^{2+}$ activity and severe wing phenotypes. These phenotypes are also observed when the p.(Asp1019Gly) or p.(Asp1165Gly) variants are overexpressed in the wing pouch, arguing that these are gain-of-function variants. However, the wing phenotypes associated with p.(His380Arg) or p.(Leu597Phe) overexpression are either mild or only partially penetrant. Our data suggest that the heterozygous missense variants reported here affect protein function differentially and contribute to the clinical features observed in the affected individuals.

## Introduction

The inositol lipid-specific phospholipase C (PLC) isozymes are key signaling proteins that play critical roles in transducing signals from hormones, growth factors, neurotransmitters, and many extracellular stimuli (*Berridge and Irvine, 1984*; *Exton, 1996*; *Balla, 2013*). The PLCs selectively hydrolyze phosphatidylinositol 4,5-bisphosphate (PIP$_2$) into inositol 1,4,5-trisphosphate (IP$_3$) and diacylglycerol (DAG) (*Nishizuka, 1984*; *Majerus et al., 1986*). PIP2 functions as a membrane anchor for numerous proteins and affects membrane dynamics and ion transport (*Hilgemann et al., 2001*; *Hilgemann, 2007*; *Suh and Hille, 2008*). The two products, IP$_3$ and DAG, are important intracellular second messengers involved in Ca$^{2+}$ signaling regulation and protein kinase C signaling activation, respectively (*Nishizuka, 1992*; *Berridge, 1993*). Hence, PLC orchestrates diverse cellular processes and behaviors, including cell growth, differentiation, migration, and cell death (*Yang et al., 2012*; *Cocco et al., 2015*; *Gomes et al., 2021*; *Asano et al., 2022*). There are at least 13 PLC isozymes grouped in 6 classes (β, δ, ε, γ, $\eta$, $\zeta$) in mammals with similar enzymatic function, but each PLC has its own spectrum of activators, expression pattern, and subcellular distribution (*Suh et al., 2008*; *Kadamur and Ross, 2013*; *Katan and Cockcroft, 2020*).

*PLCG1* [MIM: 172420] encodes the PLCγ1 isozyme. PLCγ1 can be directly activated by receptor tyrosine kinases (RTKs) as well as cytosolic receptors coupled to tyrosine kinases (*Gresset et al., 2012*). Upon tyrosine phosphorylation, PLCγ1 undergoes conformational changes that release its autoinhibition upon which it associates with the plasma membrane to bind and hydrolyze its substrates (*Gresset et al., 2010*; *Hajicek et al., 2019*; *Nosbisch et al., 2022*). There is a second PLCγ isozyme, PLCγ2, encoded by *PLCG2* [MIM: 600220]. Although these two isozymes have similar protein structure and activation mechanism, they are differentially expressed and regulated, and play non-redundant roles (*Homma et al., 1989*; *Regunathan et al., 2006*). *PLCG2* is mostly expressed in cells of the hematopoietic system and mainly functions in immune response, causing human diseases associated with immune disorders (*Yu et al., 2005*; *Ombrello et al., 2012*; *Zhou et al., 2012*; *Neves et al., 2018*; *Baysac et al., 2024*). However, *PLCG1* is ubiquitously expressed and is enriched in the central nervous system (CNS) (*GTEx Consortium, 2015*). *Plcg1* is essential in mice, and a null allele causes embryonic lethality with developmental defects in the vascular, neuronal, and immune system (*Ji et al., 1997*; *Liao et al., 2002*). *PLCG1* has emerged as a possible driver for cell proliferation, and increased expression levels of *PLCG1* have been observed in breast cancer, colon cancer, and squamous cell carcinoma (*Arteaga et al., 1991*; *Noh et al., 1994*; *Park et al., 1994*; *Xie et al., 2010*). Moreover, hyperactive somatic mutations of *PLCG1* have been observed in angiosarcomas and T cell leukemia/lymphomas

(*Behjati et al., 2014*; *Kunze et al., 2014*; *Vaqué et al., 2014*; *Kataoka et al., 2015*). However, the genotype-phenotype association of germline *PLCG1* variants has yet to be explored.

Here, we reported seven individuals carrying heterozygous variants in *PLCG1* (GenBank: NM_002660.3) who exhibit partially overlapping clinical features including hearing impairment (5/7), ocular pathology (4/7), cardiac defects (3/6), abnormal brain MRI findings (2/3), and immunological issues with diverse manifestations (5/7). Utilizing *Drosophila* to model the variants *in vivo*, we provide evidence that the missense *PLCG1* variants are toxic and affect protein function to varying degrees. We argue that these variants contribute to the clinical symptoms observed in the affected individuals.

## Results

### Individuals with heterozygous missense variants in *PLCG1* exhibit hearing impairment, cardiac defects, ocular pathology, and immune dysregulation

Seven individuals with heterozygous missense variants in *PLCG1* were recruited through the Undiagnosed Diseases Network (UDN) (*Splinter et al., 2018*) (Individuals 1–2) and GeneMatcher (*Sobreira et al., 2015*) (Individuals 3–7). Individual 1 [c.3056A>G, p.(Asp1019Gly)], Individual 2 [c.1139A>G, p.(His380Arg)] and Individual 3 [c.3494A>G, p.(Asp1165Gly)] are *de novo* cases from unrelated families. Individuals 4–7 are from the same family, and all carry the *PLCG1* variant [c.1789C>T p. (Leu597Phe)]. The phenotypes of the individuals partially overlap but show a spectrum of clinical manifestations.

Briefly, Individual 1 is an 18-year-old male who presented with multiple joint contractures, stiffness, and difficulty with gait. He was also diagnosed with pyloric stenosis, congenital mild hearing loss, bilateral posterior embryotoxon with Axenfeld anomaly, and ventricular septal defect (VSD). Brain MRI revealed stable mild diffuse cerebral and cerebellar volume loss and stable multifocal gliosis within the supratentorial white matter. Spine MRI was consistent with mild caudal regression syndrome. He had gross motor delay, partly related to multiple joint contractures since infancy.

Individual 2 is a 14-year-old female with left congenital microphthalmia and a right-sided optic nerve hypoplasia. At the age of 11 years, she presented with a slowly progressive relapsing inflammatory encephalomyelitis with optic neuritis and a leukoencephalopathy that was rapidly responsive to corticosteroid treatment but was unable to be withdrawn from the treatment. Her brain MRI showed progressive strikingly symmetric changes, consisting of primarily white matter swelling, persistent diffuse T2 FLAIR, and confluent hyperintensities within the frontoparietal lobes bilaterally. Apart from a slightly reduced NK cell CD56 level (58; normal range 76–800), there was no obvious immunodeficiency identified.

Individual 3 is a 9-year-old male with a history of intrauterine growth retardation. He had a mild to moderate sensorineural deafness from birth. He also presented with a small subaortic VSD and a wide atrial septal defect (ASD) with left-to-right shunt. He had recurrent pulmonary infections in the first year of life and had an absolute T lymphocytopenia, with normal absolute B lymphocytes. He presented with some facial dysmorphism, such as left eyelid ptosis, downturned mouth, long and flat philtrum, anteverted nostrils, prognathism, and evident veins of the scalp.

Individual 4 is an 11-year-old female. She presented with bilateral congenital and progressive sensorineural hearing loss, mild motor delay, and a unilateral posterior embryotoxon. She also had a history of joint inflammation, tarsal synovitis, recurrent respiratory and lung infections, as well as inflammatory lymphadenopathy with normal hemato-immunologic tests.

Individual 5 is the mother of Individual 4. She is 42 years old and had a severe immunologic history starting at the age of 6 years with immune thrombocytopenic purpura, polyarthritis, autoimmune pulmonary fibrosis, pneumococcal sepsis with chronic thrombocytopenia and IgA and IgG2 deficiency. She also presented with congenital ventricular septal defect and a high myopia since adolescence. She experienced sudden bilateral hearing loss during corticosteroid therapy, and by the age of 42 years, she had moderate bilateral sensorineural hearing loss.

Individual 6 is the brother of Individual 4. He is 13 years old and has a history of recurrent upper respiratory and lung infections due to a mild IgA and IgG2 deficiency. His hearing is normal.

Individual 7 is the maternal grandfather of Individual 4. He is 66 years old and has moderate bilateral hearing loss starting in adulthood with no other clinical finding.

A summary of the clinical findings is shown in *Table 1* and detailed case reports are included in Appendix 1.

## The missense *PLCG1* variants affect conserved protein domains and are predicted to be deleterious

*PLCG1* is predicted to be tolerant to loss-of-function alleles with a pLI (probability of being loss-of-function intolerant) score (*Lek et al., 2016*) of 0.16, suggesting that loss of one copy of the gene is unlikely to cause haploinsufficiency in humans, consistent with the presence of many protein truncating variants in gnomAD (*Karczewski et al., 2020*). However, the missense constraint Z score (*Lek et al., 2016*) of *PLCG1* is 3.69, suggesting that it is intolerant to missense variants. Consistently, all variants are located within regions or stretches depleted in missense variants according to scores such as regional missense constraint (RMC) (*Chao et al., 2024*) or missense tolerance ratio (MTR) (*Sun et al., 2024*). In addition, the prediction based on the DOMINO algorithm indicates that *PLCG1* variants are likely to have a dominant effect (*Quinodoz et al., 2017*). Several other in-silico pathogenicity predictions also suggest that these variants are likely to be pathogenic (*Table 2*) based on MARRVEL (*Wang et al., 2017*).

The four variants identified from the affected individuals map to different conserved protein domains of PLCγ1, and each variant affects an amino acid residue that is conserved from flies to humans (*Figure 1A and B*). The p.(Asp1019Gly) and p.(His380Arg) variants map to the catalytic core domains (X and Y regions, respectively), the p.(Asp1165Gly) variant is in the C-terminal C2 domain and the p.(Leu597Phe) variant is in the nSH2 domain. The latter is part of the PLCγ-specific regulatory array composed of a split PH domain (sPH), two Src homology 2 (nSH2 and cSH2) domains, and a Src homology 3 (SH3) domain. PLCγ1 also contains other conserved domains including an N-terminal pleckstrin homology (PH) domain and four EF hand motifs.

## The *small wing* (*sl*) is the fly ortholog of human *PLCG1*

Flies have three genes encoding PLC isozymes (*Figure 1—figure supplement 1*). Among them, *small wing* (*sl*) is predicted to be the ortholog of *PLCG1* with a DIOPT (DRSC Integrative Ortholog Prediction Tool) score of 17/18 (DIOPT version 9.0; *Hu et al., 2021*). The encoded proteins share 39% identity and 57% similarity and are composed of similar conserved domains (*Figure 1A*). The *sl* gene is also predicted to be the ortholog of *PLCG2* with a DIOPT score of 12/18. These data suggest that *sl* corresponds to two human genes encoding the PLCγ isozymes. To obtain information about the nature of the *PLCG1* variants, we utilize Drosophila to model them *in vivo* using the binary GAL4 system (*Brand and Perrimon, 1993*). We generated transgenic flies carrying the *UAS-human PLCG1* cDNAs for both the reference (*UAS-PLCG1$^{Reference}$*) and the variants (*UAS-PLCG1$^{D1019G}$*, *UAS-PLCG1$^{H380R}$*, *UAS-PLCG1$^{D1165G}$*, and *UAS-PLCG1$^{L597F}$*). Given the high level of protein sequence homology and the conservation of the affected amino acids (*Figure 1B*), we also generated transgenic flies for the reference and analogous variants in the fly *sl* cDNA, namely *UAS-sl$^{WT}$* and *UAS-sl$^{variants}$* (*UAS-sl$^{D1041G}$*, *UAS-sl$^{H384R}$*, *UAS-sl$^{D1184G}$*, and *UAS-sl$^{L630F}$*).

In Drosophila, *sl* is on the X chromosome, and several alleles of *sl* have been isolated or previously generated, including *sl$^2$*, *sl$^{KO}$*, and *sl$^{T2A}$* (*Figure 1C*). *sl$^2$* carries a 13 bp deletion in the third exon that leads to a frameshift and early stop gain (*Thackeray et al., 1998*). *sl$^2$* is a strong loss-of-function allele that causes small wing size, ectopic wing veins and extra R7 photoreceptors (*Thackeray et al., 1998*). *sl$^{KO}$* was generated by CRISPR-mediated genomic editing that removes nearly the entire gene (*Trivedi et al., 2020*). *sl$^{T2A}$* allele was generated by inserting an FRT-SA-T2A-GAL4-polyA-FRT cassette as an artificial exon into the first coding intron of *sl* (*Figure 1C*; *Diao et al., 2015*; *Lee et al., 2018*). The polyA arrests transcription, and *sl$^{T2A}$* is a strong loss-of-function allele (*Figure 1—figure supplement 2*). The T2A viral sequence triggers ribosomal skipping and leads to the production of GAL4 proteins (*Donnelly et al., 2001*; *Diao and White, 2012*) that are expressed in the proper spatial and temporal pattern of *sl*. This allows us to assess the expression pattern of *sl* by driving the expression of a *UAS-fluorescent protein* (*Lee et al., 2018*), or to assess the function of variants by expressing the human *UAS-reference/variant cDNAs* (*Huang et al., 2022a*; *Huang et al., 2022b*; *Lu et al., 2022a*; *Lu et al., 2022b*; *Ma et al., 2023*; *Pan et al., 2023*). In addition, the cassette is flanked by two FRT sites and can

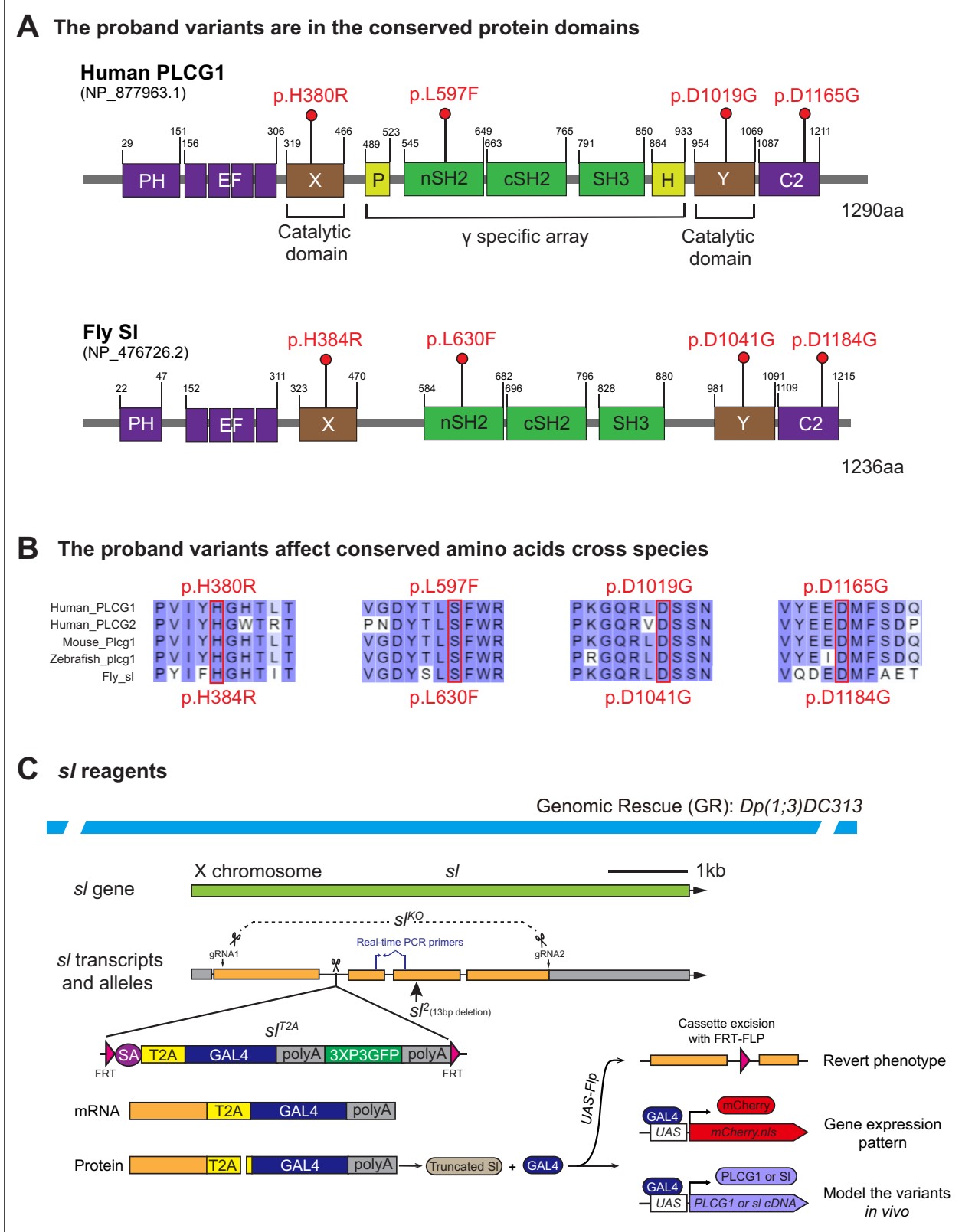

**Figure 1.** The *PLCG1* ortholog is *small wing* (*sl*) in Drosophila. (**A**) Schematic of human PLCG1 and fly Sl protein domains and positions of the variants identified in the affected individuals. Domain prediction is based on annotation from NCBI. (**B**) Alignment of protein domains near variants of PLCG1 and PLCG2 with PLCG1 from other species. The variants are marked with boxes. All the variants affect conserved amino acids (labeled in red). Isoforms for alignment: Human PLCG1 NP_877963.1; Human PLCG2 NP_002652.2; Mouse Plcg1 NP_067255.2; Zebrafish plcg1 NP_919388.1; Fly sl NP_476726.2.

*Figure 1 continued on next page*

*Figure 1 continued*

(**C**) Schematic of fly *sl* genomic span, transcript, alleles, and the 92 kb genomic rescue (GR) construct. Loss-of-function alleles of *sl* including *sl²* (13 bp deletion, **Thackeray et al., 1998**), *sl^KO* (CRISPR-mediated deletion of the gene span; **Trivedi et al., 2020**), and *sl^T2A* (T2A cassette inserted in the first intron; **Lee et al., 2018**) are indicated. The T2A cassette in *sl^T2A* is flanked by FRT sites and can be excised by Flippase to revert loss-of-function phenotypes. GAL4 expression in *sl^T2A* is driven by the endogenous *sl* promoter, allowing assessment of *sl* gene expression pattern with a *UAS-mCherry. nls* reporter line. This system also allows *in vivo* modeling of proband-associated variants by crossing with human *PLCG1* cDNAs or corresponding fly *sl* cDNAs. The primer pair used for real-time PCR is indicated.

The online version of this article includes the following source data and figure supplement(s) for figure 1:

**Figure supplement 1.** Human PLC genes and orthologs in Drosophila.

**Figure supplement 1—source data 1.** Source data for *Figure 1—figure supplement 1*.

**Figure supplement 2.** Real-time PCR reveals that *sl^T2A* is a loss-of-function allele causing severely reduced mRNA expression of *sl*.

**Figure supplement 2—source data 1.** Source data for *Figure 1—figure supplement 2*.

therefore be excised from the cells that express the gene in the presence of *UAS-Flippase* to revert the mutant phenotypes (**Figure 1C**; **Lee et al., 2018**).

We first assessed the expression pattern of *sl* by driving *UAS-mCherry.nls* (an mCherry that localizes to nuclei) with *sl^T2A*. *sl* is expressed in the 3^rd larval wing discs and eye discs (**Figure 2A**), consistent with the loss-of-function phenotypes observed in the wings and eyes (**Thackeray et al., 1998**). The expression pattern of *sl* in the wing discs is not homogenous. Higher expression levels are observed in the anterior compartment and along both the anterior/posterior and dorsal/ventral compartment boundaries (**Figure 2A**). The hemizygous *sl^T2A/Y* male flies and the trans-heterozygous *sl^T2A/sl²* or *sl^T2A/ sl^KO* female flies show reduced wing size and ectopic wing veins (**Figure 2B**, **Figure 2—figure supplement 1A**), as well as additional photoreceptors in the eye (**Figure 2C**, **Figure 2—figure supplement 1B**). These phenotypes can be rescued by *UAS-Flippase* or by introducing a genomic rescue construct (*Dp(1;3)DC313*; **Venken et al., 2010**, **Figure 1C**) that covers the *sl* locus (**Figure 2B and C**). These data show that all the observed phenotypes in *sl^T2A* mutants can be attributed to the loss of *sl*.

## The *sl* gene is expressed in the fly CNS and loss of *sl* causes longevity and locomotion defects

Given that human *PLCG1* is highly expressed in the CNS (**GTEx Consortium, 2015**) and that the affected individuals present with neurologic phenotypes including hearing or vision deficits (**Table 1**), we investigated the expression pattern and the cell type specificity of *sl* in the CNS of flies. *sl* is expressed in the larval CNS as well as the adult brain, and co-staining with the pan-neuronal marker Elav (**Robinow and White, 1991**) and glial marker Repo (**Sepp et al., 2001**) shows that *sl* is expressed in many neurons and glia cells in the CNS (**Figure 3A**). We therefore assessed the longevity and climbing of *sl^T2A* flies. Compared to the wild-type *w^1118* flies, *sl^T2A/Y* hemizygous mutant flies show a shortened lifespan and a progressively reduced climbing ability. These phenotypes can be rescued by expression of the wild-type *sl* cDNA (*sl^T2A/Y; UAS-sl^WT*; **Figure 3B**).

## Functional assays in flies indicate that the *PLCG1* variants are toxic

To assess the impact of the variants, we expressed the *sl* variant cDNAs in the *sl^T2A/Y* hemizygous mutant males (*sl^T2A/Y; UAS-sl^variants*) and compared their rescue ability with the wild-type *sl* (*sl^T2A/Y; UAS-sl^WT*). As shown in **Figure 4A** (middle panel), the *sl^T2A/Y* mutant flies (or the ones expressing a *UAS-Empty* control construct) have a slightly reduced eclosion rate, but expression of the *sl^WT* cDNA fully rescues the percentage of eclosing progeny as measured by the Mendelian ratio. In contrast, expression of *sl^L630F* (*sl^T2A/Y; UAS-sl^L630F*) reduced the percentage of hemizygous male progeny from the expected 25% to approximately 17%, while expression of *sl^H384R* causes a severe reduction in the number of eclosing flies, with only a few escapers (*sl^T2A/Y; UAS-sl^H384R*). Expression of the *sl^D1041G* or *sl^D1184G* leads to 100% lethality. These data clearly indicate that these variants are toxic but at different levels.

Since the *sl^T2A/Y; UAS-cDNA* hemizygous males lack the endogenous *sl^+*, we tested *sl^T2A/yw; UAS-cDNA* heterozygous female flies that carry a copy of wild-type *sl^+* while simultaneously expressing *UAS-cDNAs* driven by the *sl^T2A* driver in the cells that endogenously express *sl* (**Figure 4A**, right panel). The eclosion rates of heterozygous female progeny expressing *sl* variants were significantly reduced

**Table 1.** Clinical features of the affected individuals.

| | Individual 1 | Individual 2 | Individual 3 | Individual 4 | Individual 5 | Individual 6 | Individual 7 |
|---|---|---|---|---|---|---|---|
| **PLCG1 variants** | c.3056A>G p.(Asp1019Gly) | c.1139A>G p.(His380Arg) | c.3494A>G p.(Asp1165Gly) | c.1798C>T p.(Leu597Phe) | c.1798C>T p.(Leu597Phe) | c.1798C>T p.(Leu597Phe) | c.1798C>T p.(Leu597Phe) |
| **Inheritance pattern** | de novo, Sanger confirmed | de novo, Sanger confirmed | de novo, Sanger confirmed | Inherited | Inherited | Inherited | Unknown |
| **Gender** | Male | Female | Male | Female | Female | Male | Male |
| **Age at evaluation** | 18 years | 14 years | 9 years | 11 years | 42 years | 13 years | 66 years |
| **Age of onset** | Hearing loss since birth; other congenital anomalies recognized in infancy | Congenital microphthalmia/optic atrophy; episodic steroid-responsive inflammatory encephalomyelitis/optic neuritis from 11 years | Congenital hearing loss and heart defects | Congenital | 6 years | Childhood | Adulthood |
| **Developmental milestones** | Motor delays due to joint contractures; speech delay due to hearing loss | Developmental history limited; started walking at ~2 yo, articulation from 4 years | Normal | Motor delays | Normal | Normal | NA |
| **Hearing** | Mild hearing loss | Normal | Mild-moderate sensorineural hearing loss | Sensorineural bilateral, congenital progressive, profound hearing loss | Moderate bilateral sudden and progressive sensorineural hearing loss | Normal | Mild-moderate bilateral sensorineural hearing loss |
| **Vision** | Axenfeld anomaly bilaterally; posterior embryotoxon | Bilateral but variable congenital eye malformation | Normal | Unilateral posterior embryotoxon | High myopia | Normal | Normal |
| **Heart** | Cardiac septal defects (closed spontaneously) | Normal | Ventricular septal defect; atrial septal defect | Normal | Ventricular septal defect | Normal | NA |
| **Brain MRI abnormality** | Stable mild diffuse cerebral and cerebellar vermian volume loss, stable multifocal gliosis within the supratentorial white matter | Relapsing steroid-responsive inflammatory encephalomyelitis and progressive symmetrical white matter changes with swelling, and persistent diffuse T2 hyperintensities (deep and periventricular white matter), and bilateral frontoparietal lobe confluent hyperintensities | Normal | NA | NA | NA | NA |
| **Immunological issues** | No concerns reported | Symmetric steroid-responsive neuroinflammation | Lymphocytopenia (T lymphocytes), frequent infections during the first year of life | Episodes of joint inflammation, tarsal synovitis, recurrent upper respiratory and lung infections, and inflammatory lymphadenopathy Routine immunological evaluations revealed no biological abnormalities | Mild B lymphopenia, IgG2 severe deficit, splenectomy, post-vaccination sepsis septicemia, several autoimmune clinical manifestations, ITP (immune thrombocytopenic purpura) | Normal lymphocytes, IgA and IgG2 mild deficits; frequent oropharynx and lung infections | No concerns reported |
| **Skin disorders** | Multiple lentigines, keratosis pilaris | Striae seen over lower abdomen and bilateral inner thighs, possibly secondary to steroid use | Thin skin, prominent veins of the scalp | Absent | Absent | Absent | Absent |
| **Joint** | Joint stiffness/contractures; bilateral coxa profunda; trigger finger, cubitus valgus | Normal | Normal | Articular inflammations, tarsus synovitis episodes | Normal | Normal | NA |
| **Dysmorphisms** | Relative macrocephaly | Absent | Relative macrocephaly, facial dysmorphism | Absent | Absent | Absent | Absent |

*Table 1 continued on next page*

*Table 1 continued*

|  | Individual 1 | Individual 2 | Individual 3 | Individual 4 | Individual 5 | Individual 6 | Individual 7 |
|---|---|---|---|---|---|---|---|
| Short stature | Absent | Absent | Yes | Absent | Absent | Absent | Absent |
| Other potential variants | Intragenic *PSD3* duplication, paternally inherited | *ERAP2* and *SEMA3G* (compound heterozygous variants for both) | *de novo* heterozygous missense variant in *PKP2* | No | No | No | No |

Individual 1 carries an intragenic duplication in *PSD3*. *PSD3* has not been associated with a Mendelian disorder but is potentially associated with an autosomal dominant arthrogryposis (**Bayram et al., 2016**). Hence, it may underlie the joint defects observed in individual 1.

Individual 2 has compound heterozygous missense variants in *ERAP2* and *SEMA3G*. *ERAP2* [MIM: 609497] has not been associated with a Mendelian disorder. It encodes an ER-residential metalloaminopeptidase that functions in the major histocompatibility class I antigen presentation pathway. Some variants in *ERAP2* are associated with a susceptibility to autoimmune diseases such as ankylosing spondylitis and Crohn's disease (**Franke et al., 2010**; **Cortes et al., 2013**; **Ebrazeh et al., 2021**; **Venema et al., 2024**). Given that individual 2 exhibits neuroinflammation and encephalitis, these phenotypes may be associated with the *ERAP2* variants. *SEMA3G* (Semaphorin 3G) has not been associated with a Mendelian disorder. However, a homozygous missense variant in *SEMAG3* was observed in two affected siblings from a consanguineous family. The siblings exhibited dysmorphic features as well as developmental delay (**Oleari et al., 2021**).

Individual 3 carries a *de novo* missense variant in *PKP2* [MIM: 602861]. *PKP2* encodes Pakophilin-2 and has been associated with dominant arrhythmogenic right ventricular dysplasia 9 [MIM: 609040] (**Gerull et al., 2004**; **Dalal et al., 2006**; **Hakui et al., 2022**). However, this individual was born with septal defects.

compared to those expressing $sl^{WT}$. Expression of $sl^{H384R}$ or $sl^{L630F}$ in the heterozygous progeny reduced the expected 25% proportion to approximately 10% and 20%, respectively, whereas expression of $sl^{D1041G}$ or $sl^{D1184G}$ resulted in complete lethality in heterozygous flies. These results suggest that the missense variants exert a dominant toxic effect. Additionally, we observed that the toxicity may have both developmental and acute effects in adults, with varying severity among the different variants (*Figure 4—figure supplement 1*), indicating that *sl* function is required in adult flies, implying that *PLCG1* variants may cause long-term deficits in affected individuals.

To compare the *sl* and *PLCG1*-associated phenotypes, we conducted similar assays using human *PLCG1* cDNAs (*Figure 4B*). Expression of $PLCG1^{Reference}$ in the $sl^{T2A}/Y$ mutant flies ($sl^{T2A}/Y$; *UAS-$PLCG1^{Reference}$*) reduces viability by 80%, and expression of the other PLCγ coding gene, *PLCG2*, is also toxic and causes similar viability reduction compared to $PLCG1^{Reference}$. This suggests that expression of human PLCγ genes is toxic in flies. This toxicity appears to be associated with expression level (*Figure 4—figure supplement 2*), and the survivals of $sl^{T2A}/Y$; *UAS-$PLCG1^{Reference}$* did not show rescue of the loss-of-function phenotypes in the wings or eyes (*Figure 4—figure supplement 3*). Expression of $PLCG1^{H380R}$ or $PLCG1^{L597F}$ in the $sl^{T2A}/Y$ mutant flies ($sl^{T2A}/Y$; *UAS-$PLCG1^{H380R}$* or $sl^{T2A}/Y$; *UAS-$PLCG1^{L597F}$*) leads to a significant but very modest increase in lethality when compared to $PLCG1^{Reference}$, whereas expression of $PLCG1^{D1019G}$ or $PLCG1^{D1165G}$ results in 100% lethality (*Figure 4B*, left panel). When the reference and variants are assayed in the presence of a wild-type copy of $sl^+$, the heterozygous female progeny expressing the reference cDNA of *PLCG1* or *PLCG2* exhibited normal eclosion rate, as did the ones expressing $PLCG1^{H380R}$ or $PLCG1^{L597F}$, suggesting that the presence of a wild-type copy of $sl^+$ combined with the reduced expression levels (typically 50% due to dosage compensation for the genes on X chromosome) masks some of the potential toxicity. However, expression of $PLCG1^{D1019G}$ or $PLCG1^{D1165G}$ still resulted in complete lethality in the females (*Figure 4B*, right panel). In summary, expression of the *PLCG1* variants and the corresponding fly *sl* variants exhibits greater toxicity than the reference or wild-type proteins with varying degrees of severity, suggesting that the variants are likely to be gain-of-function or neomorphic alleles. Among them, the $PLCG1^{D1019G}$ and $sl^{D1041G}$, as well as $PLCG1^{D1165G}$ and $sl^{D1184G}$, are very strong toxic alleles, whereas $PLCG1^{H380R}$, $PLCG1^{L597F}$, and their fly analogues are mild variants.

## The p.(Asp1019Gly) and p.(Asp1165Gly) variants are hyperactive

To assess whether the variants act as gain-of-function alleles that enhance the enzymatic activity of the PLCγ1 isozyme, we tested them using a $Ca^{2+}$ reporter assay. Since one of the products of the PLCγ1 isozyme, $IP_3$, binds to receptors on the endoplasmic reticulum to trigger $Ca^{2+}$ release (**Foskett et al., 2007**), intracellular $Ca^{2+}$ levels can serve as a proxy of the PLCγ1 enzymatic activity. We expressed the *CaLexA* (calcium-dependent nuclear import of LexA) reporter (**Masuyama et al., 2012**) in the wing disc pouch region using a specific GAL4 driver (*nub-GAL4>UAS-CaLexA.GFP*) while simultaneously expressing *UAS-PLCG1* cDNAs. We first assessed three control variants: $PLCG1^{H380A}$, $PLCG1^{D1165H}$, and $PLCG1^{S1021F}$. Substitution of His380 with Ala (H380A) has been reported to suppress $PIP_2$ hydrolysis and $IP_3$ production (**Smith et al., 1994**; **Wada et al., 2022**), acting as an enzymatic-dead loss-of-function

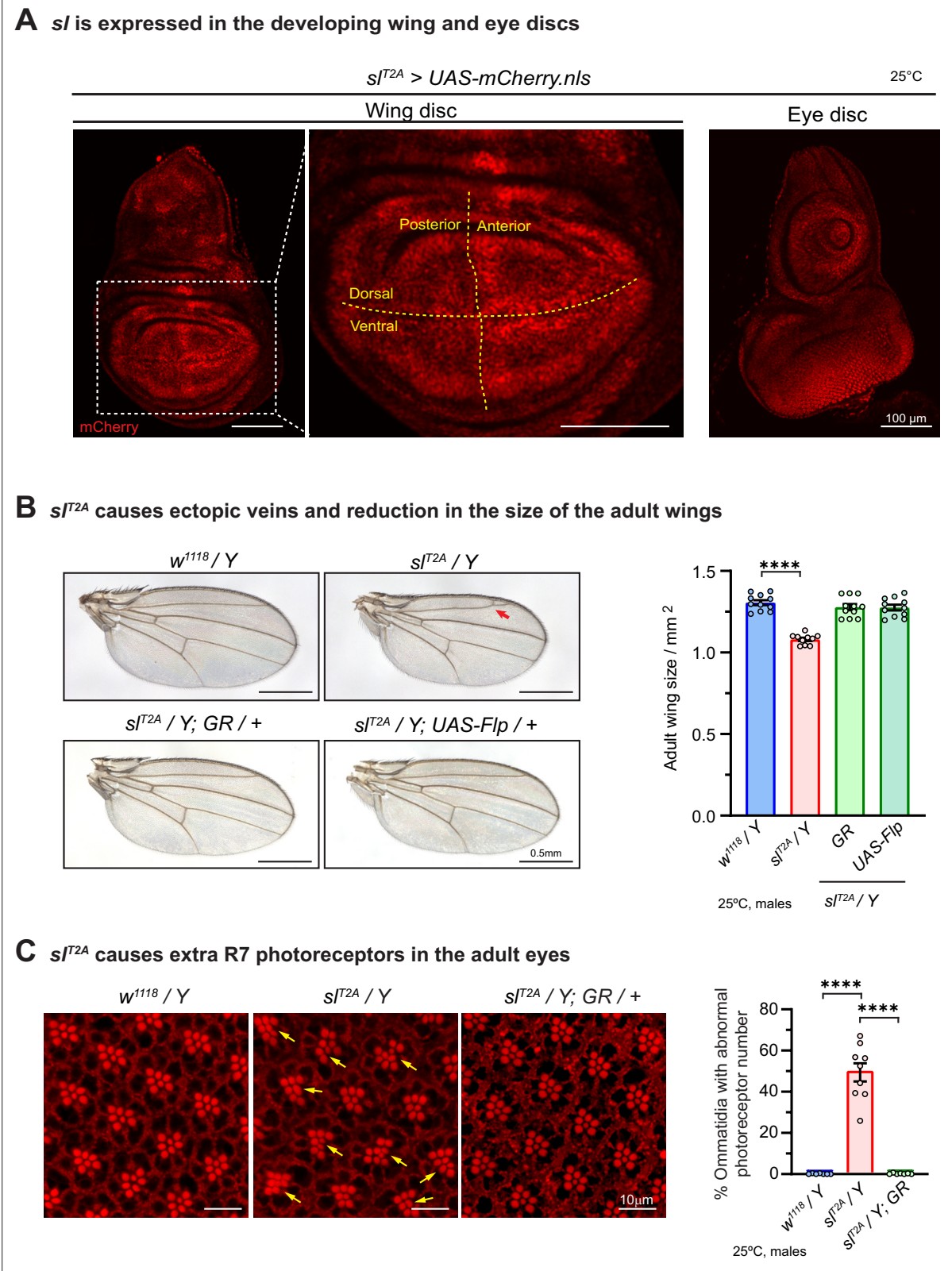

**Figure 2.** $sl^{T2A}$ is a loss-of-function allele that affects fly wing and eye development. (**A**) $sl$ expression in wing and eye discs. Expression of *UAS-mcherry.nls* (red) was driven by $sl^{T2A}$ to label the nuclei of the cells that expressed $sl$. $sl$ is expressed in the 3rd instar larval wing disc (left) and eye disc (right). A higher magnification image of the wing disc pouch region indicated the by dashed rectangle is shown. The posterior/anterior and dorsal/ventral compartment boundaries are indicated by dashed lines in yellow. Scale bars, 100 µm. (**B**) $sl^{T2A}$ cause a wing size reduction and ectopic veins (arrow) in

*Figure 2 continued on next page*

*Figure 2 continued*

hemizygous mutant male flies. The wing phenotypes can be rescued by introduction of a genomic rescue (GR) construct or the expression of Flippase. Scale bars, 0.5 mm. The quantification of adult wing size is shown in the right panel. Each dot represents the measurement of one adult wing sample. Unpaired t test, ****p<0.0001, mean ± SEM. (C) *sl^T2A* causes extra photoreceptors (arrows) in the hemizygous mutant flies. The eye phenotype can be rescued by introduction of a genomic rescue (GR) construct. The photoreceptor rhabdomeres stain positive for phalloidin labeling F-actin. Scale bars, 10 μm. The quantification is shown in the right panel. Each dot represents the measurement of one retina sample. Unpaired t test, ****p<0.0001, mean ± SEM.

The online version of this article includes the following source data and figure supplement(s) for figure 2:

**Source data 1.** Source data for *Figure 2B and C*.

**Figure supplement 1.** Trans-heterozygous *sl* mutant flies exhibit defects in wing and eye development.

allele. On the other hand, the p.(Asp1165His) (D1165H) variant was previously identified as a strong gain-of-function somatic variant in adult T cell leukemia/lymphoma (*Kataoka et al., 2015*; *Hajicek et al., 2019*; *Siraliev-Perez et al., 2022*), and has been documented to cause a dramatic increase in phospholipase activity *in vitro* (*Hajicek et al., 2019*; *Siraliev-Perez et al., 2022*). The p.(Ser1021Phe) variant was reported recently in a *de novo* case and was characterized as a gain-of-function germline variant (*Tao et al., 2023*). As shown in *Figure 5—figure supplement 1A*, the GFP signal of the *CaLexA.GFP* reporter was low in wing discs expressing *PLCG1^H380A*, whereas the signal was significantly enhanced in those expressing *PLCG1^D1165H* or *PLCG1^S1021F*, showing that this is a robust assay for detecting increased enzymatic activity. We next tested the variants of the affected individuals. As shown in *Figure 5A*, expression of *PLCG1^Reference* did not induce obvious GFP signals, suggesting that the protein is not enzymatically active, possibly because of autoinhibition. Similarly, expression of *PLCG1^H380R* or *PLCG1^L597F* did not significantly alter the GFP signal, suggesting that they are not constitutively active. However, expression of *PLCG1^D1019G* or *PLCG1^D1165G* markedly increased the GFP signal, similar to the *PLCG1^D1165H* and *PLCG1^S1021F* positive controls (*Figure 5A*, *Figure 5—figure supplement 1A*). The same observations were made with the fly *sl* variants (*Figure 5—figure supplement 1B*). These results indicate that the *PLCG1^D1019G* and *PLCG1^D1165G* variants are hyperactive, whereas the *PLCG1^H380R* and *PLCG1^L597F* variants are not hyperactive based on this assay.

## The *PLCG1* variants affect size and morphology of wings and eyes

To further assess the impact of the *PLCG1* variants on normal development, we analyzed the morphology of the adult wings upon wing-specific expression of *PLCG1* or *sl* cDNAs (*nub-GAL4>UAS-cDNAs*). Interestingly, ectopic expression of either *PLCG1^Reference* or *sl^WT* in the wing disc leads to an ~10% reduction in adult wing size when compared to the *UAS-Empty* control (*Figure 5—figure supplement 2A*). This observation, together with the reduced wing size seen in the loss-of-function context (*Figure 2B*), suggests that both reduced and elevated levels of PLCγ1 can impair wing growth. This implies a dosage-dependent regulation on wing growth by the PLCγ1 isozymes, while the underlying mechanism is unknown. Additionally, as shown in *Figure 5B*, *Figure 5—figure supplement 2B*, approximately 10% of the wings expressing *PLCG1^Reference* exhibit notching along the wing margin,

**Table 2.** Pathogenicity prediction of the proband variants.

|  | Individual 1 | Individual 2 | Individual 3 | Individual 4–7 |
|---|---|---|---|---|
| *PLCG1* variants (NM_002660.2) | c.3056A>G (p.Asp1019Gly) | c.1139A>G (p.His380Arg) | c.3494A>G (p.Asp1165Gly) | c.1789C>T (p.Leu597Phe) |
| CADD | 34 | 26.3 | 34 | 25.7 |
| M-CAP | Damaging, 0.7070 | Damaging, 0.8303 | Damaging, 0.7607 | Damaging, 0.5872 |
| PolyPhen2 hDiv (rare allele) | Probably Damaging, 0.9120 | Probably Damaging, 0.7456 | Probably Damaging, 0.9120 | Probably Damaging, 0.9058 |
| PolyPhen2 hVar (Mendelian Disease) | Probably Damaging, 0.8948 | Probably Damaging, 0.6982 | Probably Damaging, 0.9756 | Probably Damaging, 0.9737 |
| Mutation Taster | Disease Causing | Disease Causing | Disease Causing | Disease Causing |
| Count in gnomAD | Absent | Absent | Absent | Absent |

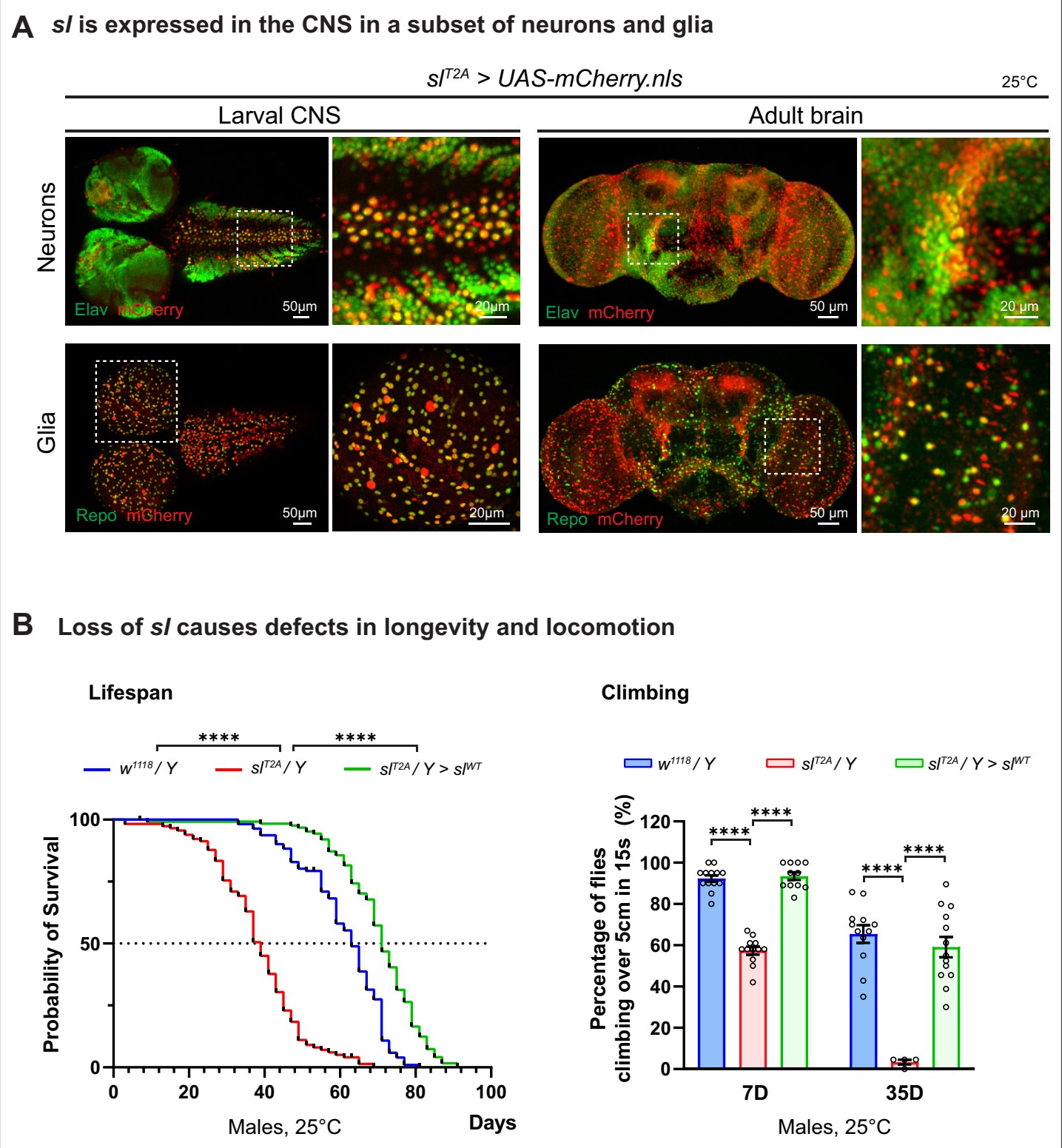

**Figure 3.** *sl* is expressed in a subset of neurons and glia in the CNS, and loss of *sl* causes behavioral defects. (**A**) Expression pattern of *sl* in the central nervous system observed by *sl^{T2A}*-driven expression of *UAS-mCherry.nls* reporter (red). In either larval or adult brain, *sl* is expressed in a subset of fly neurons and glia, which were labeled by pan-neuronal marker Elav (green, upper panel) and pan-glia marker Repo (green, lower panel). Higher magnification images of the regions indicated by dashed rectangles are shown. Scale bars, 20 µm in the magnified images, 50 µm in other images. (**B**) Loss of *sl* causes defects in longevity and locomotion. *sl^{T2A}* hemizygous flies have a shorter lifespan than *w^{1118}* control flies. The median lifespan of *sl^{T2A}* and *w^{1118}* flies is 40 days and 62 days, respectively. The shorter lifespan of *sl^{T2A}* flies can be rescued by a UAS transgene that expresses the wild-type *sl*

*Figure 3 continued on next page*

*Figure 3 continued*

cDNA (*sl^WT^*). Fly locomotion was assessed by climbing assay (see Materials and methods). *sl^T2A^* flies at the age of 7 days show reduced locomotion and become almost immotile at the age of 35 days. The reduced locomotion ability in *sl^T2A^* flies can be fully rescued by *sl^WT^*. For lifespan assay, Longrank test, ****p<0.0001; sample size n=114, 115, and 125 for *w^1118^*, *sl^T2A^*, and *sl^T2A^>sl^WT^* flies, respectively. For climbing assay, each dot represents a measurement of one vial containing 17–22 flies for test. Unpaired t test, ****p<0.0001, mean ± SEM.

The online version of this article includes the following source data for figure 3:

**Source data 1.** Source data for *Figure 3B*.

a phenotype not observed in wings expressing *sl^WT^*. Expression of *PLCG1^H380R^* or *PLCG1^L597F^* caused notched wings in approximately 18% and 23% of the flies, respectively (***Figure 5—figure supplement 2B***), whereas expression of *PLCG1^D1019G^* or *PLCG1^D1165G^* results in severe wing phenotypes characterized by notched wing margins, fused/thickened veins, and reduced wing sizes with >95% penetrance (***Figure 5B***). Notably, expression of fly *sl^variants^* could lead to similar morphological defects as their corresponding human variants, arguing that these wing phenotypes are due to alterations of PLCG1 or Sl protein function (***Figure 5B***, ***Figure 5—figure supplement 2C***).

We also assessed the effect of expression of human *PLCG1* on eye development using the *eyeless-GAL4* (*ey-GAL4*). Expression of *PLCG1^Reference^* or *PLCG1^H380R^* in fly eyes leads to a mild reduction in eye size when compared to *UAS-Empty* control (***Figure 5C***, ***Figure 5—figure supplement 2D***). However, expression of *PLCG1^L597F^* results in rough eyes that are reduced in size, whereas overexpression of *PLCG1^D1019G^* or *PLCG1^D1165G^* leads to a more severe eye phenotype (***Figure 5C***, ***Figure 5—figure supplement 2***). In summary, the eye data are consistent with the wing data, showing that *PLCG1^D1019G^* and *PLCG1^D1165G^* are more toxic than *PLCG1^Reference^*. On the other hand, the toxicity of *PLCG1^H380R^* and *PLCG1^L597F^* is stronger than the *PLCG1^Reference^* but not as severe as *PLCG1^D1019G^* and *PLCG1^D1165G^*. Interestingly, the morphological defects in wings or eyes caused by ectopic expression of *PLCG1* cDNAs correlate with the expression level (***Figure 5—figure supplement 2E***), but do not directly correlate with the phospholipase enzymatic activity. For example, expression of *PLCG1^S1021F^* does not cause obvious morphological defects when compared to *PLCG1^Reference^* (***Figure 5—figure supplement 2B and F***), even though *PLCG1^S1021F^* is hyperactive and induces significantly elevated intracellular $Ca^{2+}$ in the CaLexA reporter assay (***Figure 5—figure supplement 1A***).

## Discussion

Here, we report seven individuals who carry heterozygous missense variants in *PLCG1* which encodes the phospholipase C γ1 isozyme. The individuals present with partially overlapping clinical features including hearing impairment, eye abnormality, heart defects, and immune phenotypes. We show that the fly ortholog, *small wing* (*sl*), is widely expressed in wings and eyes, as well as in the central nervous system. Consistent with its expression pattern, we report that *sl* not only regulates wing and eye development, as previously documented, but also plays critical roles in the nervous system and affects locomotion and longevity. Furthermore, we assessed the function of the variants in the context of the human and fly cDNAs and show that their expression induces variable levels of toxicity when compared to the reference *PLCG1* or wild-type *sl*. Two of the variants are clearly hyperactive, and all the variants exhibit neomorphic effects (discussed in Appendix 1 as Figure Notes). These observations show that the variants impair the normal function *in vivo* and suggest that they contribute to the symptoms observed in the affected individuals. Similarly to inborn error caused by the paralogous *PLCG2* (***Baysac et al., 2024***), germline variants in *PLCG1* can be pathogenic and dominant by different mechanisms.

### Structural analysis of the PLCG1 variants

Previously, studies based on biochemical assays and protein structures provided insights into how the variants studied here may affect the enzymatic activity of PLCγ1 (the protein structure of full-length rat Plcg1 is shown in ***Figure 6A***). In its basal state, the PLCγ-specific regulatory array (sPH-nSH2-cSH2-SH3) forms autoinhibitory interfaces with the catalytic domains. Upon activation by the RTKs through binding with nSH2, PLCγ1 is phosphorylated, which induces the dissociation of the inhibitory cSH2 domain from the C2 domain. This triggers conformational rearrangements, allowing the enzyme to associate with the membrane and to expose the catalytic domains to allow hydrolysis of

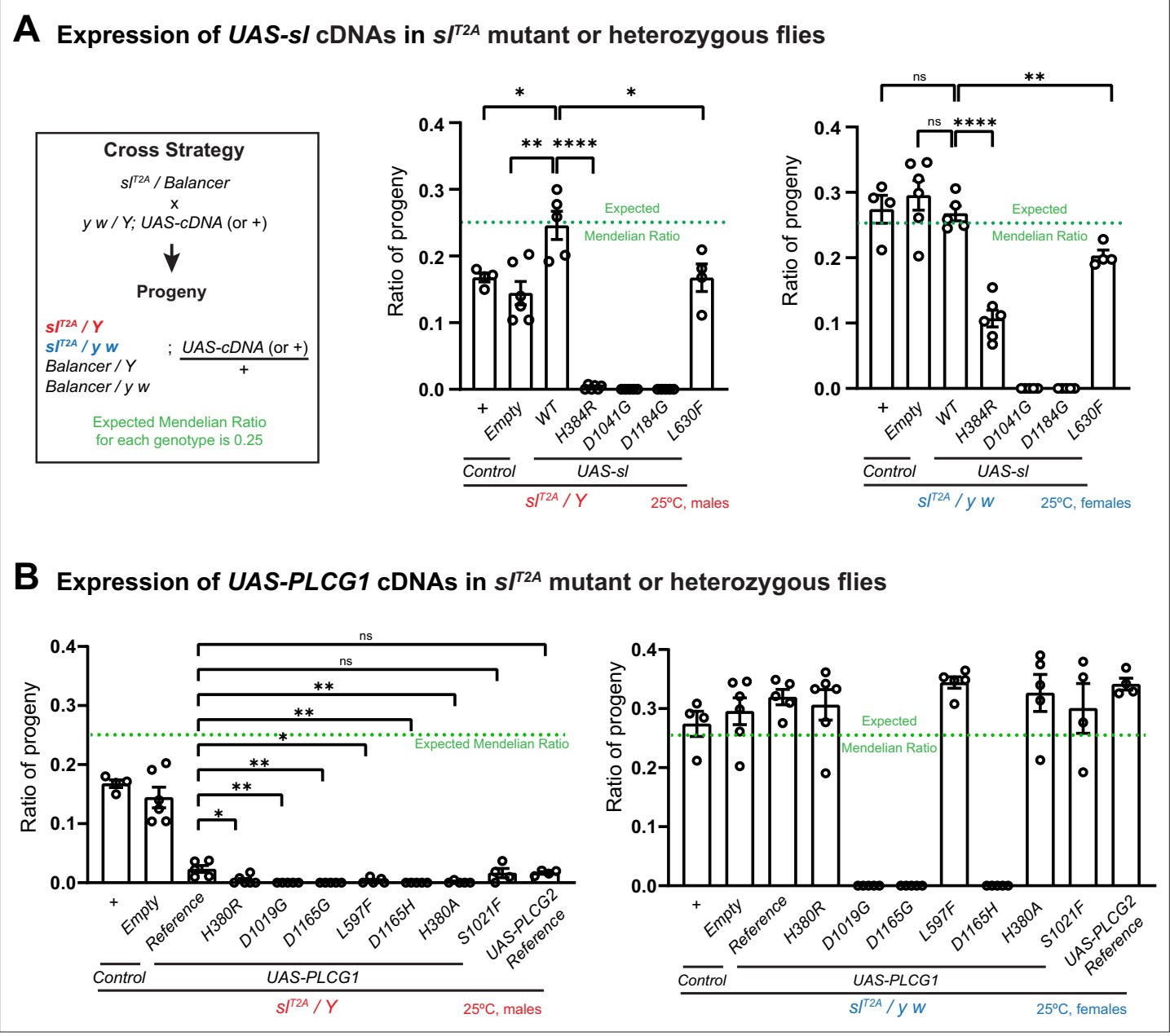

**Figure 4.** The human and corresponding fly variants are toxic when expressed in flies. (**A**) Summary of the viability associated with expression of *sl* cDNAs in *sl^{T2A}* mutant or heterozygous flies. Cross strategy: heterozygous *sl^{T2A}* female flies were crossed to male flies carrying *UAS-cDNAs* or control (*UAS-Empty*) constructs, or crossed to the *y w* males as an extra control. The percentages of hemizygous *sl^{T2A}/Y* male progeny (red) or *sl^{T2A}/yw* heterozygous female progeny (blue) that express different *UAS-cDNA* constructs were calculated. The expected Mendelian ratio is 0.25 (indicated by the green line in the graph). The fly analogue variants of the proband-associated variants were tested. Each dot represents one independent replicate. Unpaired t test, ****p<0.0001, **p<0.01, *p<0.05, ns: not significant, mean ± SEM. (**B**) Summary of the viability associated with the expression of *PLCG1* cDNAs in *sl^{T2A}* mutant (red, males) or heterozygous (blue, females) flies. The same cross strategy and progeny ratio measurement described in (**A**) was applied. The proband-associated variants, as well as three previously reported *PLCG1* variants were assessed. We also included the *PLCG2* reference cDNA. Each dot represents one independent replicate. Unpaired t test, **p<0.01, *p<0.05, ns: not significant, mean ± SEM.

The online version of this article includes the following source data and figure supplement(s) for figure 4:

**Source data 1.** Source data for *Figure 4A and B*.

**Figure supplement 1.** Proband-associated variants exhibit deleterious impacts in adults.

**Figure supplement 1—source data 1.** Source data for *Figure 4—figure supplement 1B and C*.

**Figure supplement 2.** The toxicity of human *PLCG1* cDNAs in flies correlates with expression levels.

*Figure 4 continued on next page*

*Figure 4 continued*

**Figure supplement 2—source data 1.** Source data for *Figure 4—figure supplement 2A–C*.

**Figure supplement 3.** Expressing human *PLCG1* does not rescue wing or eye phenotypes associated with *sl^{T2A}*.

**Figure supplement 3—source data 1.** Source data for *Figure 4—figure supplement 3*.

PIP2 (*Gresset et al., 2010*; *Hajicek et al., 2019*; *Liu et al., 2020*; *Le Huray et al., 2022*; *Nosbisch et al., 2022*). As shown in *Figure 6A*, the proband-associated variants map to conserved domains of the protein, either within the catalytic domains or at intramolecular and intermolecular interfaces. The p.(Asp1019Gly) and p.(Asp1165Gly) variants impact key residues involved in autoinhibition, leading to increased enzymatic activity. Specifically, the p.(Asp1019Gly) variant affects a conserved residue within the hydrophobic ridge of the Y box (*Figure 6B*), which is important for interaction with the sPH domain. This interaction is critical for the autoinhibition by blocking the membrane engagement of the catalytic core domain prior to enzymatic activation (*Ellis et al., 1998*; *Hajicek et al., 2019*). Notably, a substitution at the same position (Asp1019Lys, D1019K) has been demonstrated to enhance basal phospholipase activity *in vitro* (*Hajicek et al., 2013*), supporting its regulatory importance. Similarly, another hotspot somatic variant, p.(Ser345Phe), located in the corresponding hydrophobic ridge within the X box, is also hyperactive (*Vaqué et al., 2014*; *Manso et al., 2015*). On the other hand, the p.(Asp1165Gly) variant affects a residue situated within a loop of the C2 domain (*Figure 6C*). The Asp1165 residue plays a key role in stabilizing the interaction between the cSH2 domain and the C2 domain to maintain the autoinhibited state (*DeBell et al., 2007*). As mentioned above, the somatic variant p.(Asp1165His) leads to significantly elevated phospholipase activity *in vitro* (*Liu et al., 2020*; *Siraliev-Perez et al., 2022*), and results in severe phenotypes *in vivo* (*Figure 4B*, *Figure 5—figure supplement 2E and F*). Molecular dynamics simulation data consistently indicate that autoinhibition is likely disrupted by the p.(Asp1019Gly) and p.(Asp1165Gly) variants (*Figure 6—figure supplement 1A*). In contrast, the p.(His380Arg) variant impacts the His380 residue within the X box, situated near a $Ca^{2+}$ ion in the catalytic core (*Figure 6D*). His380 plays a role in coordination of the phosphate group at the 1-position of $IP_3$ (*Le Huray et al., 2022*). While this residue may not be key to autoinhibition, it is important for the phospholipase activity. Substitution of His380 with Phe or Ala (H380F, H380A) has been reported to suppress $PIP_2$ hydrolysis and $IP_3$ production (*Smith et al., 1994*; *Wada et al., 2022*). Hence, substitution of the His380 with Arg by the p.(His380Arg) variant may create a more basic environment, impacting the lipase activity. On the other hand, the p.(Leu597Phe) variant affects a residue within the nSH2 domain, which is part of the PLCγ-specific regulatory array (*Figure 6E*). The nSH2 domain mediates interactions with phosphorylated tyrosine residues on RTKs to initiate activation (*Bae et al., 2009*). Leu597 is located near the phosphotyrosine-binding pocket, and this variant may therefore alter receptor specificity or induce novel protein interactions. Additionally, we utilized the DDMut platform (*Zhou et al., 2023*) to predict protein stability and folding of the variants, which are discussed in *Figure 6—figure supplement 1B*. In summary, our *in vivo* data are consistent with previous reports and *in silico* analyses, showing that the affected amino acids map to critical residues and strengthening the conclusion that the variants are pathogenic and likely impact the protein function through distinct mechanisms.

## The *PLCG1* variants affect protein function to varying degrees and are associated with variable clinical manifestations

To better assess the genotype-phenotype relationship of the variants, we summarize the clinical features of affected individuals in *Table 1*, and the phenotypic effects observed in fly assays in *Table 3*. The p.(Asp1019Gly) variant carried by Individual 1 and the p.(Asp1165Gly) variant carried by Individual 3, and their corresponding fly variants induce severe phenotypes across all assays performed. Individuals 1 and 3 share several obvious clinical features including hearing loss and heart septal defect. In contrast, the p.(His380Arg) and p.(Leu597Phe) variants cause mild or partially penetrant phenotypes across different fly assays. Individual 2 who carries the p.(His380Arg) variant does not exhibit hearing impairment or heart defects observed in Individuals 1 and 3, but has eye malformations and neuroinflammation features that are shared with individuals 1 and 3, although the ocular and immunological defects manifest differently among individuals. Interestingly, individuals 4–7 are from the same family

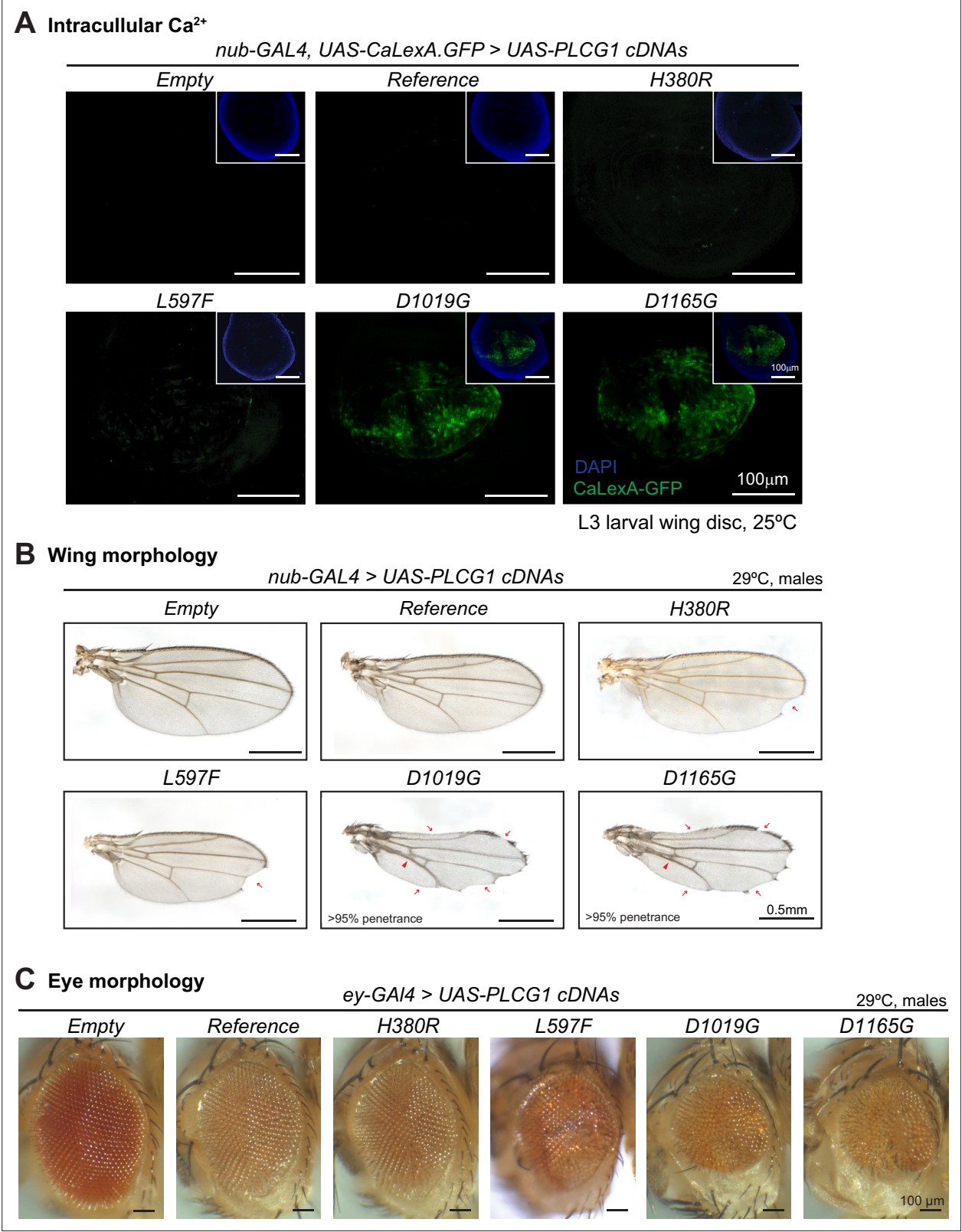

**Figure 5.** Ectopic expression of *PLCG1* variants causes variable phenotypes. (**A**) The $Ca^{2+}$ reporter *CaLexA.GFP* was expressed in the wing disc pouch, simultaneously with the *PLCG1* cDNAs. Expression of *PLCG1^{D1019G}* or *PLCG1^{D1165G}* caused elevated CaLexA.GFP signal (green), indicating increased intracellular $Ca^{2+}$ levels, indicating that these variants are hyperactive. Nuclei were labeled with DAPI (blue). Scale bars, 100 μm. (**B**) Representative images of the adult wing blades showing the morphological phenotypes caused by wing-specific expression of *PLCG1* cDNAs. Expression of

*Figure 5 continued on next page*

*Figure 5 continued*

$PLCG1^{D1019G}$ or $PLCG1^{D1165G}$ caused severe wing morphology defects including notched margin (arrows) and fused/thickened veins (arrowheads). Expression of $PLCG1^{L597F}$ exhibited partial penetrance. Expression of $PLCG1^{H380R}$ exhibited very mild phenotypes, comparable to $PLCG1^{Reference}$. Scale bars, 0.5 mm. (**C**) Representative images showing that eye-specific expression of $PLCG1^{Reference}$ or $PLCG1^{H380R}$ causes an ~15% eye size reduction compared to the *UAS-Empty* control construct, and expression of $PLCG1^{L597F}$ further reduced eye size. Expression of $PLCG1^{D1019G}$ or $PLCG1^{D1165G}$ causes a severe size reduction by ~30%. Scale bars, 100 µm.

The online version of this article includes the following source data and figure supplement(s) for figure 5:

**Figure supplement 1.** Intracellular Ca$^{2+}$ reporter assay suggests that the p.(Asp1019Gly) and p.(Asp1165Gly) are hyperactive variants.

**Figure supplement 2.** Wing and eye phenotypes associated with ectopic expression of *PLCG1* variants.

**Figure supplement 2—source data 1.** Source data for *Figure 5—figure supplement 2A–C*.

and all carry the p.(Leu597Phe) variant but also differ in their phenotypes, yet all share some clinical features with Individuals 1–3 (*Table 1*).

The heterogeneity in clinical manifestations may be influenced by additional genetic variants (see *Table 1* legend) and environmental factors. Additionally, the variable expressivity observed in carriers of the same variant may be explained by allelic expression bias through autosomal random monoallelic expression (aRME; *Reinius and Sandberg, 2015*), a phenomenon that is thought to be common among carriers of genetic defects associated with inborn errors of immunity (IEIs). Indeed, these conditions often exhibit non-Mendelian segregation patterns and variable clinical features (*Stewart et al., 2025*). Moreover, the PLCγ1 isozyme is an integral component of multiple signaling pathways, and the consequences of its dysregulation are likely to be context dependent. It is likely that different *PLCG1* variants impact distinct cellular processes across various tissues and cell types, resulting in a spectrum of pathological changes. In summary, the symptoms observed in affected individuals appear to correlate, to some extent, with the severity of the variants as indicated by fly assays. However, the penetrance and expressivity of these phenotypes will require further investigation to better understand the genotype-phenotype associations of *PLCG1* variants.

## Materials and methods
### Recruitment of the probands

Individuals 1 and 2 were recruited through the Undiagnosed Diseases Network (UDN) and were evaluated through the clinical research protocol of the National Institutes of Health Undiagnosed Diseases (15-HG-0130), which was approved by the National Human Genome Research Institute (NHGRI). Individuals 3–7 were recruited through GeneMatcher. Formal consent for genetic testing and participation, as well as for publication under the Creative Commons Attribution 4.0 International Public License (CC BY 4.0), was obtained from all individuals or their family members.

### Drosophila husbandry and generation of transgenic flies

All the flies used in this study were raised and maintained on standard fly food at room temperature unless specified. The *UAS-PLCG1 cDNAs* and *UAS-sl cDNAs* transgenic flies were generated in-house (see Materials and methods below). Other fly strains used in this study were obtained from the Bloomington *Drosophila* Stock Center (BDSC), including: $w^{1118}$ (#3605), $y\ w$ (#1495), $sl^2$ (#5735), $sl^{KO}$ (#93748), $sl^{T2A}$ (#81213), *Dp(1;3)DC313* (Genomic Rescue) (#30423), *UAS-Flippase* (#4539), *UAS-mCherry.nls* (#38424), *tub-GAL4* (#5138), *da-GAL4* (#55851), *nub-GAL4* (#86108), *ey-GAL4* (#5534), *tub-GAL80ts* (#7107), *UAS-Empty* (#9750). The $sl^{T2A}$ allele was outcrossed with $w^{1118}$ to clean up the genetic background.

To generate the *UAS-cDNA* transgenic lines, human *PLCG1* cDNA was obtained from Horizon Discovery (MHS6278-213246131, clone ID 9052656), and fly *sl* cDNA was obtained from Drosophila Genomics Resource Center (DGRC, RE62235). The coding sequence (CDS) of $PLCG1^{Reference}$ and $sl^{WT}$ was amplified using iProof High-Fidelity DNA Polymerase Kit (BioRad, #1725301), purified using QIAEX II Gel Extraction Kit (QIAGEN, #20021), sub-cloned into the Gateway compatible entry vector pDONR223 by BP cloning (BP clonase II, Thermo Fisher Scientific, #11789020) and sequentially cloned into the destination vector pGW-attB-HA by LR cloning (LR clonase II, Thermo Fisher Scientific, #11791100) (*Bischof et al., 2013*). The variants were generated by site-directed mutagenesis strategy

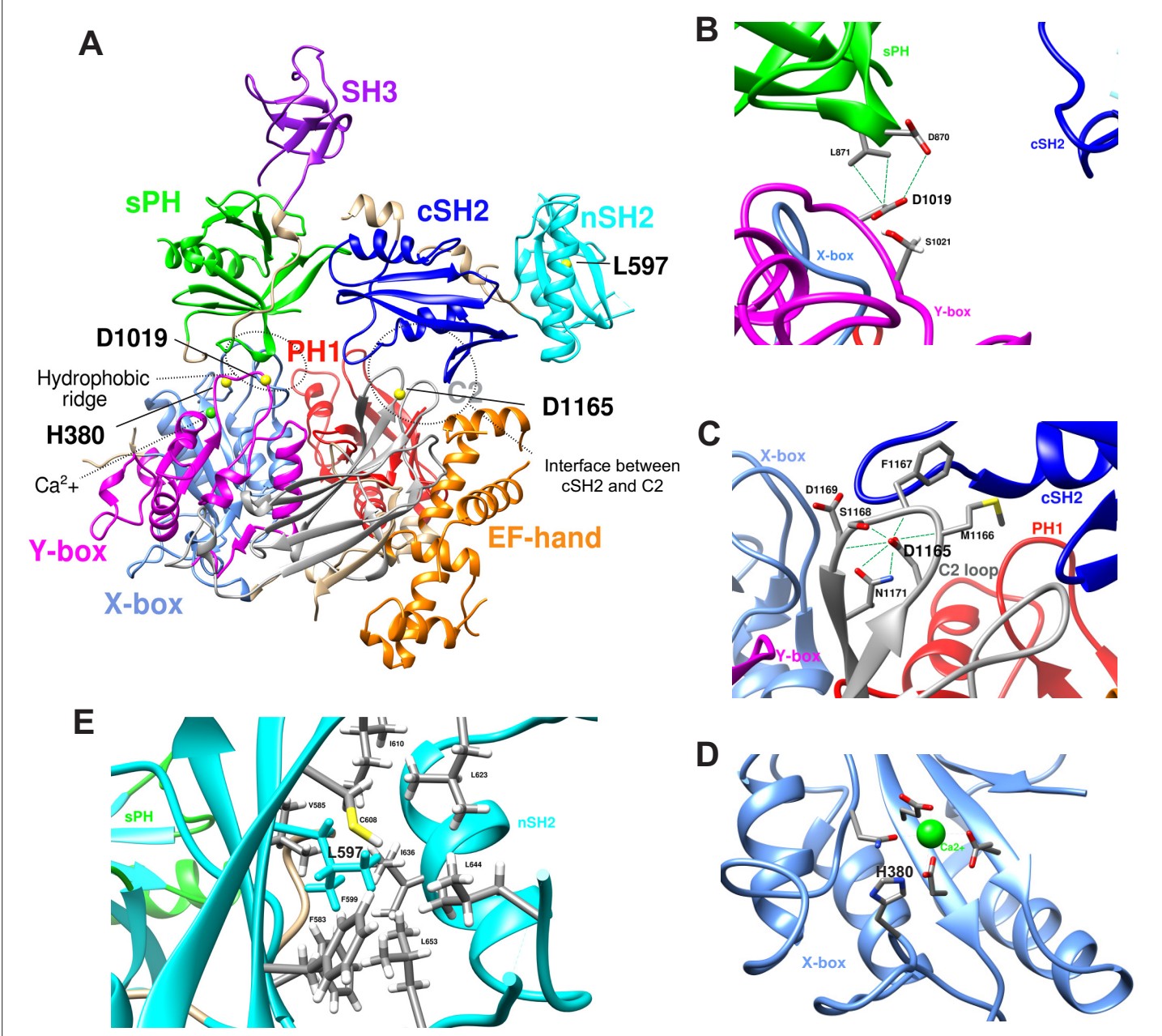

**Figure 6.** *PLCG1* variants affect important residues. (**A**) 3D structure of full-length rat Plcg1 (rat Plcg1 shares 97% amino acid identity with human PLCG1). The conserved protein domains are labeled with different colors. Two major intracellular interfaces are circled by dashed lines: 1-The hydrophobic ridge between the sPH domain and the catalytic core (X-box and Y-box); and 2-The interface between the cSH2 domain and the C2 domain. The four amino acids affected by the variants are shown as bolded black and indicated by yellow balls. (**B**) Enlarged views of the Asp1019 residue within the autoinhibition interface between sPH domain and the Y box. The potential interactions with nearby residues are indicated. (**C**) Enlarged view of the Asp1165 residue within the autoinhibition interface between the cSH2 domain and the C2 domain. The potential interactions with nearby residues are indicated. (**D**) Enlarged view of the His380 residue within the X-box catalytic domain, in proximity to the Ca²⁺ cofactor. (**E**) Enlarged view of the Leu597 and nearby residues in the nSH2 domain. Structural analysis was performed via UCSF Chimera (*Pettersen et al., 2004*).

The online version of this article includes the following figure supplement(s) for figure 6:

**Figure supplement 1.** *In silico* analyses of *PLCG1* variants.

using Q5 Hot Start High-Fidelity 2x Master Mix (NEB, #M0494S) and *DpnI* restriction enzyme (NEB, # R0176L). Human *PLCG2* cDNA was obtained from Genescript (clone ID OHu24072) and was cloned into pUAST vector using *NotI* and *XbaI* restriction enzyme sites. All the constructs were Sanger verified and injected and inserted into the VK33 (PBac{y[+]-attP}VK00033) docking site using φC31 mediated transgenesis (*Venken et al., 2006*; *Bischof et al., 2007*). Primers are listed in *Supplementary file 1*.

## Drosophila behavioral assays

For the lifespan assay, newly eclosed male flies were collected and maintained at 25°C (10 flies per vial). The flies were transferred to a new vial and the number of dead flies was counted every 2 days.

For the temperature-shifting related assays, flies were raised at 18°C until eclosion. Newly eclosed males were collected and maintained at 29°C for the lifespan assay and climbing assays conducted at specified ages.

The climbing assay to examine the negative geotaxis and locomotion ability of the flies was performed as previously described (*Madabattula et al., 2015*; *Lu et al., 2022a*) with some modifications. For climbing assay of $sl^{T2A}$ mutant, $sl^{WT}$ rescue, and $w^{1118}$ control flies, 17–22 flies per vial were transferred to an empty plastic vial and given 20 min to rest prior to being tested. The flies were tapped to the bottom of the vial and were allowed to climb for 15 s. The percentage of flies per vial that climbed over 5 cm was calculated. For climbing assay of the flies underwent temperature-shifting, the distance each fly climbed in 15 s were measured. The maximum distance from the bottom to the top is 18.5 cm.

## Immunostaining

Fly tissues were dissected in 1x PBS, fixed in 4% paraformaldehyde for 20 min at room temperature, and washed in PBS (3 x 10 min). For antibody staining, samples were treated with PBST (Triton X-100 in PBS, 0.1% for larval tissues, 2% for adult brain), 5% normal goat serum, and incubated in primary antibody overnight at 4°C. Samples were washed with 0.1% PBST (3 x 10 min) and incubated with secondary antibody for 2 hr at room temperature (in darkness) and washed in 0.1% PBST (3 x 10 min). Primary antibodies: rat anti-Drosophila Elav (1:250, DSHB, #7E8A10); mouse anti-Drosophila Repo (1:50, DSHB, #8D12). Secondary antibodies: goat anti-rat-647 (1:250, Jackson ImmunoResearch, #112-605-003), goat anti-mouse-Cy5 (1:250, Invitrogen, #A10524). Larval discs were mounted in Vectashield (Vector Labs #H1200 and #H1000). Larval CNS and adult brain were mounted in Rapiclear (Cedarlane, #RC147001). For adult retinas, flies are reared at 25°C under 12 hr light/dark conditions. Retinas were isolated from 5- to 7-day-old flies. Heads were dissected in PBS and fixed in 3.7% formaldehyde overnight at 4°C. The samples were rinsed with 0.1% PBST, and the retinas were subsequently dissected and incubated with PBST-diluted phalloidin 647 (1:100, Invitrogen, #A22287) for

**Table 3.** Summary of the phenotypes observed in fly assays.

| | 25°C | | | | | | 29°C | | |
|---|---|---|---|---|---|---|---|---|---|
| | Lethality when expressed in $sl^{T2A}$/Y mutant | | Lethality when expressed in $sl^{T2A}$/y w heterozygous | | Ca²⁺ activity | | Wing morphology when overexpressed | | Eye morphology when overexpressed |
| *PLCG1* variants | Human variants | Fly variants | Human variants | Fly variants | Human variants | Fly variants | Human variants | Fly variants | Human variants |
| Reference | ++ | - | - | - | - | - | + | + | + |
| H380R | +++ | +++ | - | ++ | - | - | ++ | ++ | + |
| D1019G | 100% lethal | 100% lethal | 100% lethal | 100% lethal | + | + | ++++ | ++++ | +++ |
| D1165G | 100% lethal | 100% lethal | 100% lethal | 100% lethal | + | + | ++++ | lethal | +++ |
| L597F | +++ | ++ | - | + | - | - | +++ | +++ | ++ |
| H380A | +++ | NA | - | NA | - | NA | (+) | NA | - |
| D1165H | 100% lethal | NA | 100% lethal | NA | + | NA | lethal | NA | lethal |
| S1021F | ++ | NA | - | NA | + | NA | + | NA | - |

'-': no obvious phenotypes observed.

'+': phenotypes observed, the number of '+' corresponds to the severity of the observed phenotype.

NA: Not Available.

1 hr. Retinas were washed in 0.1% PBST and mounted in Vectashield. The images were obtained with a confocal microscope (Leica SP8X or Zeiss Airyscan LSM 880) and processed using the ImageJ-FIJI software (*Schneider et al., 2012*).

## Imaging of adult fly wings and eyes

To prepare the samples of adult fly wings, the wing blades were dissected and mounted in a glycerol/ethanol 1/1 mixture. Only wings from the same gender were compared to each other since females have larger wings than males when raised in the same conditions. To prepare the samples of adult fly eyes, the flies were frozen and placed onto a double-sided stick tape with one eye facing up. The samples were imaged using bright field Stereomicroscope (Leica MZ16 or Leica Z16 APO). Image Pro Plus 7.0 software was used to create extended depth-of-field images. The image processing and the measurement of the total areas of the wing blades or eyes were conducted using the ImageJ-FIJI software (*Schneider et al., 2012*).

## Real-time PCR

Real-time PCR was performed as previously described (*Ravenscroft et al., 2020*) with modifications. All-In-One 5X RT MasterMix (abm, #G592), iTaq Universal SYBR Green Master Mix (BioRad, #1725120) and BioRad C1000 Touch Cycler were used. Primers are listed in *Supplementary file 1*.

## Molecular dynamics simulations

The three-dimensional structures of the wild-type and variant forms of the PLCG1 protein were predicted using AlphaFold3 (*Abramson et al., 2024*). All simulations were performed using GROMACS (*Pronk et al., 2013*) version 2020.6. Initial PDB files were processed to remove water molecules and hydrogen atoms. The AMBER14SB_parmbsc1 force field (*Maier et al., 2015*) was employed for parameterization. TIP3P water model was used to solvate the system in a cubic box with a minimum distance of 1.0 nm between the protein and box edges. Counterions ($Na^+$ and $Cl^-$) were added to neutralize the system's net charge. Energy minimization was conducted using the steepest descent algorithm to eliminate unfavorable contacts. Subsequently, the system underwent equilibration in two phases: NVT Equilibration: Maintained at 300 K using the velocity-rescaling thermostat for 100 ps; NPT Equilibration: Pressure was stabilized at 1 bar using the Parrinello-Rahman barostat for 100 ps.

An unrestrained production MD simulation was carried out for 100 ns under constant temperature (300 K) and pressure (1 bar) conditions. The LINCS algorithm was used to constrain all bonds involving hydrogen atoms, allowing a time step of 2 fs. Long-range electrostatics were treated using the Particle Mesh Ewald (PME) method with a cutoff of 1.0 nm for both Coulomb and van der Waals interactions.

Post-simulation analyses included root mean square deviation (RMSD) calculations to assess structural stability and radius of gyration (Rg) to evaluate compactness. All analyses were performed using built-in GROMACS tools. The results are plotted by the ggplot2 R package.

## Acknowledgements

We thank the individuals and families for their participation in this study. We thank Ms Hongling Pan for helping in the generation of transgenic fly lines. We thank Dr Meisheng Ma for suggestions about protein structure interpretation. We thank Dr Zhandong Liu for providing computational resources for performing the molecular dynamics simulations. We thank the Bloomington Drosophila Stock Center (BDSC) for providing stocks. H J B receives support from the Jan and Dan Duncan Neurological Research Institute at Texas Children's Hospital, the Huffington Foundation, as well as grants from the National Institute of Neurological Disorders and Stroke (NINDS, U54 NS093793) and the National Institutes of Health Office of the Director (R24 OD031447). This work was also supported by the Undiagnosed Diseases Network funded by grants from the National Human Genome Research Institute (NHGRI) and NINDS (U01 HG010233, U01 NS134355, U01 HG007709, U01 HG007942). Sequence data analysis was supported by the University of Washington Center for Rare Disease Research (UW-CRDR) and grants from NHGRI (U01 HG011744, UM1 HG006493, U24 HG011746). Confocal microscopy was performed in the Baylor College of Medicine Intellectual and Developmental Disabilities Research Center (IDDRC) Neurovisualization Core, supported by the Eunice Kennedy Shriver National Institute of Child Health and Human Development (NICHD, U54 HD083092). The content of this paper

is the sole responsibility of the authors and does not necessarily represent the official views of the National Institutes of Health

## Additional information

### Competing interests

Jill A Rosenfeld: Receives revenue from clinical genetic testing completed at Baylor Genetics Laboratories. Hugo J Bellen: Reviewing editor, eLife. The other authors declare that no competing interests exist.

### Funding

| Funder | Grant reference number | Author |
|---|---|---|
| Huffington Foundation | | Hugo J Bellen |
| Office of the Director | R24 OD031447 | Hugo J Bellen |
| National Institute of Neurological Disorders and Stroke | U54 NS093793 | Hugo J Bellen |
| National Human Genome Research Institute | U01 HG010233 | Undiagnosed Diseases Network |
| National Institute of Neurological Disorders and Stroke | U01 NS134355 | Undiagnosed Diseases Network |
| National Human Genome Research Institute | U01 HG007709 | Undiagnosed Diseases Network |
| National Human Genome Research Institute | U01 HG007942 | Undiagnosed Diseases Network |

The funders had no role in study design, data collection and interpretation, or the decision to submit the work for publication.

### Author contributions

Mengqi Ma, Yiming Zheng, Conceptualization, Resources, Data curation, Formal analysis, Investigation, Visualization, Writing – original draft, Writing – review and editing; Mingxi Deng, Data curation, Formal analysis, Visualization, Writing – review and editing; Shenzhao Lu, Formal analysis, Investigation, Visualization, Writing – review and editing; Xueyang Pan, Visualization, Writing – review and editing; Xi Luo, Jill A Rosenfeld, Formal analysis, Writing – review and editing; Michelle Etoundi, Investigation; David Li-Kroeger, Resources, Investigation, Visualization, Writing – review and editing; Kim C Worley, Lindsay C Burrage, Lauren S Blieden, Giuseppe Merla, Barbara Mandriani, Shinya Yamamoto, Michael F Wangler, Resources; Aimee Allworth, Pierre Blanc, Formal analysis; Wei-Liang Chen, Debdeep Dutta, Jingheng Chen, Writing – review and editing; Catherine E Otten, Data curation; Ian A Glass, Paolo Prontera, Sandrine Marlin, Seema R Lalani, Conceptualization, Resources, Data curation, Writing – review and editing; Elizabeth Blue, Data curation, Writing – review and editing; Jeremie Rosain, Conceptualization, Resources, Writing – review and editing; Hugo J Bellen, Conceptualization, Resources, Data curation, Supervision, Funding acquisition, Writing – review and editing

### Author ORCIDs

Mengqi Ma https://orcid.org/0000-0001-7345-575X
Shenzhao Lu https://orcid.org/0000-0003-3117-3900
Xueyang Pan https://orcid.org/0000-0003-4453-4971
Xi Luo https://orcid.org/0000-0002-1941-2318
David Li-Kroeger https://orcid.org/0000-0001-6473-7691
Jill A Rosenfeld https://orcid.org/0000-0001-5664-7987
Debdeep Dutta https://orcid.org/0000-0003-0989-7013
Shinya Yamamoto https://orcid.org/0000-0003-2172-8036
Hugo J Bellen https://orcid.org/0000-0001-5992-5989

## Ethics

Human subjects: Individuals 1 and 2 were recruited through the Undiagnosed Diseases Network (UDN) and were evaluated through the clinical research protocol of the National Institutes of Health Undiagnosed Diseases (15-HG-0130), which was approved by the National Human Genome Research Institute (NHGRI). Individuals 3-7 were recruited through GeneMatcher. Formal written consent for genetic testing and participation, as well as for publication under the Creative Commons Attribution 4.0 International Public License (CC BY 4.0), was obtained from all individuals or their family members.

Reviewer #2 (Public review): https://doi.org/10.7554/eLife.95887.3.sa1
Author response https://doi.org/10.7554/eLife.95887.3.sa2

## Additional files

### Supplementary files
MDAR checklist
Supplementary file 1. Primers used in this study.

### Data availability
All data generated or analysed during this study are included in the manuscript and supporting files; source data files have been provided. All reagents developed in this study are available upon reasonable request.

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

# Appendix 1

## Case Reports

### Individual 1

Individual 1 is an 18-year-old male with motor difficulties and multiple congenital anomalies. He failed his newborn hearing screen and was diagnosed with mild congenital hearing loss. He was found to have pyloric stenosis around 2 months of age. He was subsequently diagnosed with multiple joint contractures, Axenfeld-Rieger syndrome with posterior embryotoxon of both eyes, and cardiac septal defects that closed spontaneously. He had chronic joint pain, most severe in his hips. He was found to have bilateral coxa profunda. His motor milestones were delayed due to multiple joint contractures. His speech was also delayed, likely related to mild deafness. No cognitive problems were reported. On physical exam, his weight was 81.2 kg (88th percentile), height was 168.4 cm (15th percentile), and head circumference was 58.4 cm (90th percentile). He had limitation of small joints of the hands and elbows with cubitus valgus. He had limited ability to extend arms above his head. Deep tendon reflexes were normal. His tone was normal in his lower extremities. His muscle strength was normal, although mildly reduced in his hands. His gait was mildly abnormal but significantly improved after his hip surgery. Brain MRI showed stable mild diffuse cerebral and cerebellar vermian volume loss. There was no evidence of neurodegenerative process or progressive cerebellar atrophy process. Spinal MRI showed very mild caudal regression syndrome and Scheuermann's phenomenon in the midthoracic spine. Reanalysis of the trio whole exome sequencing (WES) identified a *de novo* heterozygous missense variant in *PLCG1*. Optical genome mapping identified an intragenic duplication in *PSD3* duplication, and whole genome sequencing (WGS) was performed to characterize this variant. He also had an *L1CAM* variant reported but a healthy male cousin carries this variant as well, which suggests that the *L1CAM* variant may not be a key player. Chromosomal microarray analysis showed a copy number loss within chromosome band 15q15.3 spanning approximately 0.048 Mb, involving *STRC* and *CATSPER2*, indicating that he was probably a carrier for the autosomal recessive deafness-infertility syndrome [MIM: 611102].

### Individual 2

Individual 2 is a 14-year-old female with a history of congenital microphthalmia and blindness of the right eye leading to enucleation, and several episodes of an undifferentiated inflammatory encephalopathy from age of 11 years. She was born at 38 weeks with a right microphthalmia and no useful vision. This anomaly was managed by enucleation at 2 years old with prosthesis placement. She passed her newborn hearing screen. Developmentally, she walked at 2 years old and began talking at 4 years old. She came to our medical attention at the age of 11 years with episodes of inflammatory encephalopathy, which first presented as fever with associated upper respiratory symptoms that later developed into lingering episodes of headache, fatigue, dizziness, and weight loss. Brain MRI imaging showed progressive but symmetrical primarily white matter changes with swelling, particularly of the cerebellum, persistent diffuse T2 hyperintensities (both deep and periventricular white matter), and confluent hyperintensities within the frontoparietal lobes bilaterally. Thus far, this episodic relapsing disorder has been responsive to steroids and intravenous immunoglobulin with relapse on steroid withdrawal. On physical exam, she was noted to have difficulty getting up from the floor without using support from her arms, in addition to weakness of plantar flexion, possibly secondary to chronic steroid administration. Family history is noncontributory, with both parents' ancestry being Egyptian and no consanguinity reported. Clinical genetic workup included WES, which revealed compound heterozygous variants in *RYR1*, thought to be noncontributory to the phenotype. She was accepted into the Undiagnosed Diseases Network research study in 2021 to pursue additional genetic workup. WGS revealed the genetic findings mentioned in *Table 1*: *PLCG1 de novo* variant, *ERAP2* compound heterozygous variants, and *SEMA3G* compound heterozygous variants, all Sanger confirmed.

### Individual 3

Individual 3 is a 9-year-old male born from healthy non-consanguineous parents. During pregnancy, a diagnosis of intrauterine growth retardation was made. At birth, at 39 weeks of gestation, his length was 49 cm (50° centile), weight was 2930 g (25–50° centile) and occipital frontal circumference was 34 cm (50° centile). Newborn audiological screening and the subsequent auditory brainstem response test showed a mild to moderate sensorineural deafness. Echocardiogram revealed a small subaortic ventricular septal defect and a wide atrial septal defect with left to right shunt. For the recurrence of pulmonary infections in the first year of life, he underwent a complete immunological

assessment. The immunophenotypic characterization identified an absolute T lymphocytopenia. He had a normal absolute B lymphocyte count and normal IgG subclasses. He had relative macrocephaly and presented with some facial dysmorphisms, such as left eyelid ptosis, downslanting both corners of the mouth, evident venous reticulum of the skull, long and flat philtrum, prominent ears, anteverted nostrils, prognathism, proximally placed thumbs, and evident veins of the scalp. His brain MRI was normal, as was the ophthalmological evaluation with fundus analysis. Trio WES revealed *de novo* heterozygous missense variants in *PKP2* and *PLCG1*.

## Individuals 4–7

Individual 4 is an 11-year-old female. She failed her newborn hearing screening and was diagnosed with bilateral congenital sensorineural hearing loss, which has since progressed to profound bilateral hearing impairment. She was also diagnosed with unilateral posterior embryotoxon. She experienced mild motor delay, beginning to walk at 18 months of age, despite normal vestibular test results. On physical examination, her weight, height, and head circumference were within normal ranges. Her medical history includes multiple episodes of joint inflammation, tarsal synovitis, recurrent upper respiratory and lung infections, and inflammatory lymphadenopathy. Her most recent hematologic and immunologic investigations were within normal range. The family history revealed relevant findings in her mother (Individual 5), her brother (Individual 6), and her maternal grandfather (Individual 7).

Her mother (Individual 5) has a history of severe immunologic disease, beginning at the age of 6 years with idiopathic thrombocytopenic purpura (ITP) (platelet count <2,000 /mm³), treated with corticosteroids and intravenous immunoglobulin therapy for six years, followed by a splenectomy after recurrence of thrombocytopenia. At the age of 20 years, she developed polyarthritis with positive autoimmune antibodies. Immunosuppressive treatment was ineffective and discontinued due to a severe Epstein-Barr virus (EBV) infection. She later developed autoimmune pulmonary fibrosis and multiple episodes of pancytopenia triggered by viral infections. At the age of 33 years, an IgA and IgG2 deficiency was identified. She subsequently experienced pneumococcal sepsis with purpura fulminans. Vaccinations against pneumococcus and meningococcus failed to induce an immune response. At the age of 40 years, she began intravenous immunoglobulin therapy every 3 weeks and takes regular oral antibiotics for recurrent pulmonary infections. Regarding hearing, she experienced an episode of sudden bilateral hearing loss while on corticosteroid therapy. Her sensorineural hearing loss has gradually progressed to moderate bilateral deafness by age of 42 years. She also has a congenital ventricular septal defect and high myopia since adolescence. Her growth parameters are normal, and she does not present with any neurodevelopmental disorders. The 13-year-old brother (Individual 6) has normal hearing. His growth parameters and neuromotor development are within normal ranges. Since early childhood, he had numerous rhinopharyngeal and lung infections. Recent laboratory investigations reveal a mild IgA and IgG2 deficiency. He does not exhibit any joint or ophthalmological symptoms. The maternal grandfather (Individual 7), now 66 years old, presents with moderate bilateral hearing loss that began in adulthood, without any other clinical manifestations upon history-taking. Whole genome sequencing in the 4 individuals revealed a probably pathogenic *PLCG1* variation inherited from Individual 7 (Sanger analysis).

## Notes for *Figure 4*

We assessed if human *PLCG1* could effectively serve as a functional substitute for fly *sl* and rescue the loss-of-function phenotypes observed in *sl* mutant flies. However, as shown in *Figure 4B*, only a small fraction of the *sl*[T2A]/Y mutant hemizygotes expressing *PLCG1*[Reference] can survive to adults, and the escapers die within 1 week. Since overexpression of fly *sl*[WT] does not cause viability issues (*Figure 4A*), the reduced viability associated with *PLCG1*[Reference] in *sl*[T2A] mutant male progeny may be due to elevated expression levels of the human proteins.

To assess the 'high expression level toxicity' hypothesis, we raised the flies at different temperatures and tested with various GAL4 drivers. The GAL4-UAS system is highly temperature-dependent since the promoter in the UAS construct contains an Hsp-70 promoter (*Fischer et al., 1988*), and the expression levels increase with higher temperatures and decrease with lower temperatures (*Nagarkar-Jaiswal et al., 2015*). We assessed the viability of the progeny with expression of *PLCG1* or *sl* cDNA in hemizygous mutant flies (*sl*[T2A]/Y>*UAS-cDNAs*) at 22°C, the survival ratio was increased, but the increase was subtle (*Figure 4—figure supplement 2A*). To avoid the dosage compensation effect that alters the expression level of *sl*[T2A] between males or females, we used a ubiquitous

GAL4 driver, *Tub-GAL4*, to ectopically drive the expression of *PLCG1*[Reference] ubiquitously at different temperatures. As shown in *Figure 4—figure supplement 2B*, the *Tub-GAL4>UAS-PLCG1*[Reference] flies exhibited a high lethality ratio when raised at 29°C (~63% lethal), and the surviving flies die within 1 week. This lethality ratio decreased to ~43% when the flies were raised at 25°C, whereas >90% of the flies were able to eclose as adults at 22°C. This shows that the toxicity is highly dependent on protein levels. Indeed, the expression levels can be further lowered by using a weak ubiquitous GAL4 driver, *da-GAL4*. The *da-GAL4>UAS-PLCG1*[Reference] flies were viable at the three tested temperatures (29°C, 25°C, and 22°C), but the enclosed adults at 29°C mostly died within 1 week. *da-GAL4>UAS-PLCG1*[H380R] or *UAS-PLCG1*[L597F] exhibited similar phenotypes. In contrast, *da-GAL4>UAS-PLCG1*[D1019G] or *UAS-PLCG1*[D1165G] flies were lethal at 29°C, semi-lethal (~80%–85% lethal) at 25°C, and viable at 22°C (*Figure 4—figure supplement 2C*). The animals that escape lethality at 25°C have smaller pupae (*Figure 4—figure supplement 2D*) and a reduced adult body size, arguing that growth is impeded. In summary, these data support the hypothesis that elevated expression levels of the human *PLCG1* is toxic in flies.

In addition to the toxicity, *PLCG1*[Reference] fails to rescue the phenotypes observed in the wings or eyes of the *sl* mutant flies (*Figure 4—figure supplement 3*), which is fully rescued by fly *sl*[WT] (*Figure 2*). Expression of the *PLCG2* cDNA in *sl*[T2A] mutant hemizygous males exhibited similar toxicity (*Figure 4B*) and limited ability to rescue the phenotypes caused by loss of *sl*. This suggests that despite their high DIOPT scores, the two human genes encoding PLCγ isozymes cannot fully substitute for the fly PLCγ ortholog. It is possible that during the course of evolution, *PLCG1* has acquired more specialized functions. For example, an essential step for enzymatic activation of mammalian PLCγ is the binding of its nSH2 domain to specific phosphotyrosines on the RTKs through a specific binding motif (*Songyang et al., 1993*). However, Thackeray et al. found that the consensus motif is absent in the intracellular domain of the Drosophila EGF receptor homolog DER (*Thackeray et al., 1998*), one of the three RTKs in Drosophila which is required for wing vein differentiation and photoreceptor formation (*Dickson and Hafen, 1994*; *Freeman, 1996*; *Schweitzer and Shilo, 1997*). Nevertheless, expression of the *PLCG1* variant cDNAs leads to more severely reduced eclosion rate compared to expression of the reference cDNA (*Figure 4B*), suggesting that they are detrimental variants.

## Notes for *Figure 5*

To better understand whether the phenotypes caused by the variants in fly models are associated with the enzymatic activity of the PLCγ1 isozyme, we characterized the phenotypes of transgenic flies expressing control constructs: *PLCG1*[H380A] (enzymatic dead), *PLCG1*[D1165H] (hyperactive) and *PLCG1*[S1021F] (hyperactive). Ectopic expression of the enzymatic-dead *PLCG1*[H380A] or hyperactive *PLCG1*[S1021F] in wings or eyes did not cause obvious morphological abnormalities, whereas expression of the hyperactive *PLCG1*[D1165H] caused very severe phenotypes (*Figure 5—figure supplement 2F*). Overexpression of *PLCG1*[D1165H] in the eyes or wings causes lethality at 29°C, arguing that it is highly toxic. These flies survive when they are raised at 25°C, yet the wings show severe morphological defects, including notched wing margins, thickened veins as well as reduced wing size (*Figure 5—figure supplement 2E*). These phenotypes are similar to, but more severe than, the defects observed in the wings expressing *PLCG1*[D1019G] or *PLCG1*[D1165G] (*Figure 5B*). Notably, expression of *PLCG1*[H380R] or *PLCG1*[L597F], which did not exhibit a hyperactive effect in the *CaLexA* reporter assay compared to *PLCG1*[S1021F] (*Figure 5A* and *Figure 5—figure supplement 1A*), caused a partially penetrant wing notching phenotype (*Figure 5B* and *Figure 5—figure supplement 2B*). Additionally, *PLCG1*[L597F] expression led to a reduction in eye size (*Figure 5C* and *Figure 5—figure supplement 2D*). These observations suggest that the morphological phenotypes in wings and eyes are not directly correlated with the enzymatic activity of the PLCγ1 isozyme, but may instead be associated with neomorphic effects. Similarly, the increased lethality observed in *sl*[T2A]-driven expression of cDNAs may also be associated with neomorphic effect. As shown in *Figure 4B*, when driven by *sl*[T2A], *PLCG1*[H380A] led to reduced viability in hemizygous mutant flies but not in heterozygous ones, whereas *PLCG1*[D1165H] caused 100% lethality in both genotypes. However, the hyperactive *PLCG1*[S1021F] did not show distinguishable phenotypes compared to *PLCG1*[Reference] in this assay. In summary, the *PLCG1* variants identified in this study exhibit neomorphic effects with variable degrees of severity, whereas the p.(Asp1019Gly) and p.(Asp1165Gly) variants are hyperactive and also cause neomorphic phenotypes.

## The Undiagnosed Diseases Network Consortia (Version 3.31.25)

Alyssa A. Tran, Arjun Tarakad, Ashok Balasubramanyam, Brendan H. Lee, Carlos A. Bacino, Daryl A. Scott, Elaine Seto, Gary D. Clark, Hongzheng Dai, Hsiao-Tuan Chao, Ivan Chinn, James P. Orengo, Jennifer E. Posey, Jill A. Rosenfeld, Kim Worley, Lindsay C. Burrage, Lisa T. Emrick, Lorraine Potocki, Monika Weisz Hubshman, Richard A. Lewis, Ronit Marom, Seema R. Lalani, Shamika Ketkar, Tiphanie P. Vogel, William J. Craigen, Jared Sninsky, Lauren Blieden, Sandesh Nagamani, Hugo J. Bellen, Michael F. Wangler, Oguz Kanca, Shinya Yamamoto, Christine M. Eng, Patricia A. Ward, Pengfei Liu, Adeline Vanderver, Cara Skraban, Edward Behrens, Gonench Kilich, Kathleen Sullivan, Kelly Hassey, Ramakrishnan Rajagopalan, Rebecca Ganetzky, Vishnu Cuddapah, Anna Raper, Daniel J. Rader, Giorgio Sirugo, Vaidehi Jobanputra, Allyn McConkie-Rosell, Kelly Schoch, Mohamad Mikati, Nicole M. Walley, Rebecca C. Spillmann, Vandana Shashi, Alan H. Beggs, Calum A. MacRae, David A. Sweetser, Deepak A. Rao, Edwin K. Silverman, Elizabeth L. Fieg, Frances High, Gerard T. Berry, Ingrid A. Holm, J. Carl Pallais, Joan M. Stoler, Joseph Loscalzo, Lance H. Rodan, Laurel A. Cobban, Lauren C. Briere, Matthew Coggins, Melissa Walker, Richard L. Maas, Susan Korrick, Jessica Douglas, Cecilia Esteves, Emily Glanton, Isaac S. Kohane, Kimberly LeBlanc, Rachel Mahoney, Shamil R. Sunyaev, Shilpa N. Kobren, Brett H. Graham, Erin Conboy, Francesco Vetrini, Kayla M. Treat, Khurram Liaqat, Lili Mantcheva, Stephanie M. Ware, Breanna Mitchell, Brendan C. Lanpher, Devin Oglesbee, Eric Klee, Filippo Pinto e Vairo, Ian R. Lanza, Kahlen Darr, Lindsay Mulvihill, Lisa Schimmenti, Queenie Tan, Surendra Dasari, Abdul Elkadri, Brett Bordini, Donald Basel, James Verbsky, Julie McCarrier, Michael Muriello, Michael Zimmermann, Adriana Rebelo, Carson A. Smith, Deborah Barbouth, Guney Bademci, Joanna M. Gonzalez, Kumarie Latchman, LéShon Peart, Mustafa Tekin, Nicholas Borja, Stephan Zuchner, Stephanie Bivona, Willa Thorson, Herman Taylor, Rakale C. Quarells, Ayuko Iverson, Bruce Gelb, Charlotte Cunningham-Rundles, Eric Gayle, Joanna Jen, Louise Bier, Mafalda Barbosa, Manisha Balwani, Mariya Shadrina, Rachel Evard, Saskia Shuman, Susan Shin, Andrea Gropman, Barbara N. Pusey Swerdzewski, Camilo Toro, Colleen E. Wahl, Donna Novacic, Ellen F. Macnamara, John J. Mulvihill, Maria T. Acosta, Precilla D'Souza, Valerie V. Maduro, Ben Afzali, Ben Solomon, Cynthia J. Tifft, David R. Adams, Elizabeth A. Burke, Francis Rossignol, Heidi Wood, Jiayu Fu, Joie Davis, Leoyklang Petcharet, Lynne A. Wolfe, Margaret Delgado, Marie Morimoto, Marla Sabaii, MayChristine V. Malicdan, Neil Hanchard, Orpa Jean-Marie, Wendy Introne, William A. Gahl, Yan Huang, Andrew Stergachis, Danny Miller, Elisabeth Rosenthal, Elizabeth Blue, Elsa Balton, Emily Shelkowitz, Eric Allenspach, Fuki M. Hisama, Gail P. Jarvik, Ghayda Mirzaa, Ian Glass, Kathleen A. Leppig, Katrina Dipple, Mark Wener, Martha Horike-Pyne, Michael Bamshad, Peter Byers, Runjun Kumar, Seth Perlman, Sirisak Chanprasert, Virginia Sybert, Wendy Raskind, Nitsuh K. Dargie, Chun-Hung Chan, Dr. Francisco Bustos velasq, Isum Ward, Jason Schend, Jennifer Morgan, Megan Bell, Miranda Leitheiser, Mohamad Saifeddine, Paul Berger, Rachel Li, Taylor Beagle, Alexander Miller, Beatriz Anguiano, Beth A. Martin, Brianna Tucker, Chloe M. Reuter, Devon Bonner, Elijah Kravets, Hector Rodrigo Mendez, Holly K. Tabor, Jacinda B. Sampson, Jason Hom, Jennefer N. Kohler, Jennifer Schymick, John E. Gorzynski, Jonathan A. Bernstein, Kevin S. Smith, Laura Keehan, Laurens Wiel, Matthew T. Wheeler, Meghan C. Halley, Mia Levanto, Page C. Goddard, Paul G. Fisher, Rachel A. Ungar, Raquel L. Alvarez, Sara Emami, Shruti Marwaha, Stephen B Montgomery, Suha Bachir, Tanner D Jensen, Taylor Maurer, Terra R. Coakley, Euan A. Ashley, Ali Al-Beshri, Anna Hurst, Brandon M Wilk, Bruce Korf, Elizabeth A Worthey, Kaitlin Callaway, Martin Rodriguez, Tammi Skelton, Tarun KK Mamidi, Andrew B. Crouse, Jordan Whitlock, Mariko Nakano-Okuno, Matthew Might, William E. Byrd, Albert R. La Spada, Changrui Xiao, Elizabeth C. Chao, Eric Vilain, Jose Abdenur, Kirsten Blanco, Maija-Rikka Steenari, Rebekah Barrick, Richard Chang, Sanaz Attaripour, Suzanne Sandmeyer, Tahseen Mozaffar, Alden Huang, Andres Vargas, Bianca E. Russell, Brent L. Fogel, Esteban C. Dell'Angelica, George Carvalho, Julian A. Martínez-Agosto, Layal F. Abi Farraj, Manish J. Butte, Martin G. Martin, Naghmeh Dorrani, Neil H. Parker, Rosario I. Corona, Stanley F. Nelson, Yigit Karasozen, Aaron Quinlan, Alistair Ward, Ashley Andrews, Corrine K. Welt, Dave Viskochil, Erin E. Baldwin, John Carey, Justin Alvey, Laura Pace, Lorenzo Botto, Nicola Longo, Paolo Moretti, Rebecca Overbury, Russell Butterfield, Steven Boyden, Thomas J. Nicholas, Matt Velinder, Gabor Marth, Pinar Bayrak-Toydemir, Rong Mao, Monte Westerfield, Brian Corner, John A. Phillips III, Kimberly Ezell, Lynette Rives, Rizwan Hamid, Serena Neumann, Ashley McMinn, Joy D. Cogan, Thomas Cassini, Alex Paul, Dana Kiley, Daniel Wegner, Erin McRoy, Jennifer Wambach, Kathy Sisco, Patricia Dickson, F. Sessions Cole, Dustin Baldridge, Jimann Shin, Lilianna Solnica-Krezel, Stephen C. Pak, Timothy Schedl, Allen Bale, Carol Oladele, Caroline Hendry, Emily Wang, Hua Xu, Hui Zhang, Lauren Jeffries, María José Ortuño Romero, Mark Gerstein, Michele Spencer-Manzon, Monkol Lek, Nada Derar, Odelya Kaufman, Shrikant Mane, Teodoro Jerves Serrano, Vasilis Vasiliou, Winston Halstead, Yong-Hui Jiang

