## [Editor Report · eLife Assessment]

This **important** study reveals how Drosophila may be used to investigate the role of missense variants in the PLCG1 phospholipase gene in human diseases. The experimental evidence is **compelling** and brings together rigorous analysis of clinical and model organism phenotypes with a structural analysis of the PLCG1 protein.

---

## [Referee Report · Reviewer #2 (Public review)]

The manuscript by Ma et al. reports the identification of three unrelated people who are heterozygous for de novo missense variants in PLCG1, which encodes phospholipase C-gamma 1, a key signaling protein. These individuals present with partially overlapping phenotypes, including hearing loss, ocular pathology, cardiac defects, abnormal brain imaging results, and immune defects. None of the patients present with all of the above phenotypes. PLCG1 has also been implicated as a possible driver for cell proliferation in cancer.

The three missense variants found in the patients result in the following amino acid substitutions: His380Arg, Asp1019Gly, and Asp1165Gly. PLCG1 (and the closely related PLCG2) have a single Drosophila ortholog called small wing (sl). sl-null flies are viable but have small wings with ectopic wing veins and supernumerary photoreceptors in the eye. As all three amino acids affected in the patients are conserved in the fly protein, in this work Ma et al. tested whether they are pathogenic by expressing either reference or patient variant fly or human genes in Drosophila and determining the phenotypes produced by doing so.

Expression in Drosophila of the variant forms of PLCG1 found in these three patients is toxic; highly so for Asp1019Gly and Asp1165Gly, much more modestly for His380Arg. Another variant, Asp1165His which was identified in lymphoma samples and shown by others to be hyperactive, was also found to be toxic in the Drosophila assays. However, a final variant, Ser1021Phe, identified by others in an individual with severe immune dysregulation, produced no phenotype upon expression in flies.

Based on these results, the authors conclude that the PLCG1 variants found in patients are pathogenic, producing gain-of-function phenotypes through hyperactivity. In my view, the data supporting this conclusion are robust, despite the lack of a detectable phenotype with Ser1021Phe, and I have no concerns about the core experiments that comprise the paper.

Fig. 6, the last in the paper, provides information about PLCG1 structure and how the different variants would affect it. It shows that His380, Asp1019 and Asp1165 all lie within catalytic domains or intramolecular interfaces, and that variants in the latter two affect residues essential for autoinhibition. It also shows that Ser1021 falls outside the key interface occupied by Asp1019, but more could have been said about the potential effects of Ser1021Phe.

Overall, I believe the authors fully achieved the aims of their study. The work will have a substantial impact because it reports the identification of novel disease-linked genes, and because it further demonstrates the high value of the Drosophila model for finding and understanding gene-disease linkages.

Comments on revisions:

The single recommendation I made on the original version, which was to further examine H380 mutants, has been satisfactorily addressed in the revised version.

---

## [Author Response]

The following is the authors’ response to the original reviews.

**Reviewer #1 (Public Review):**
Summary:This manuscript provides an initial characterization of three new missense variants of the PLCG1 gene associated with diverse disease phenotypes, utilizing a Drosophila model to investigate their molecular effects in vivo. Through the meticulous creation of genetic tools, the study assesses the small wing (sl) phenotype - the fly's ortholog of PLCG1 - across an array of phenotypes from longevity to behavior in both sl null mutants and variants. The findings indicate that the Drosophila PLCG1 ortholog displays aberrant functions. Notably, it is demonstrated that overexpression of both human and Drosophila PLCG1 variants in fly tissue leads to toxicity, underscoring their pathogenic potential in vivo.Strengths:The research effectively highlights the physiological significance of sl in Drosophila. In addition, the study establishes the in vivo toxicity of disease-associated variants of both human PLCG1 and Drosophila sl.Weaknesses:The study's limitations include the human PLCG1 transgene's inability to compensate for the Drosophila sl null mutant phenotype, suggesting potential functional divergence between the species. This discrepancy signals the need for additional exploration into the mechanistic nuances of PLCG1 variant pathogenesis, especially regarding their gain-of-function effects in vivo.Overall:The study offers compelling evidence for the pathogenicity of newly discovered disease-related PLCG1 variants, manifesting as toxicity in a Drosophila in vivo model, which substantiates the main claim by the authors. Nevertheless, a deeper inquiry into the specific in vivo mechanisms driving the toxicity caused by these variants in Drosophila could significantly enhance the study's impact.
**Reviewer #2 (Public Review):**
The manuscript by Ma et al. reports the identification of three unrelated people who are heterozygous for de novo missense variants in PLCG1, which encodes phospholipase C-gamma 1, a key signaling protein. These individuals present with partially overlapping phenotypes including hearing loss, ocular pathology, cardiac defects, abnormal brain imaging results, and immune defects. None of the patients present with all of the above phenotypes. PLCG1 has also been implicated as a possible driver for cell proliferation in cancer.The three missense variants found in the patients result in the following amino acid substitutions: His380Arg, Asp1019Gly, and Asp1165Gly. PLCG1 (and the closely related PLCG2) have a single Drosophila ortholog called small wing (sl). sl-null flies are viable but have small wings with ectopic wing veins and supernumerary photoreceptors in the eye. As all three amino acids affected in the patients are conserved in the fly protein, in this work Ma et al. tested whether they are pathogenic by expressing either reference or patient variant fly or human genes in Drosophila and determining the phenotypes produced by doing so.Expression in Drosophila of the variant forms of PLCG1 found in these three patients is toxic; highly so for Asp1019Gly and Asp1165Gly, much more modestly for His380Arg. Another variant, Asp1165His which was identified in lymphoma samples and shown by others to be hyperactive, was also found to be toxic in the Drosophila assays. However, a final variant, Ser1021Phe, identified by others in an individual with severe immune dysregulation, produced no phenotype upon expression in flies.Based on these results, the authors conclude that the PLCG1 variants found in patients are pathogenic, producing gain-of-function phenotypes through hyperactivity. In my view, the data supporting this conclusion are robust, despite the lack of a detectable phenotype with Ser1021Phe, and I have no concerns about the core experiments that comprise the paper.Figure 6, the last in the paper, provides information about PLCG1 structure and how the different variants would affect it. It shows that the His380, Asp1019, and Asp1165 all lie within catalytic domains or intramolecular interfaces and that variants in the latter two affect residues essential for autoinhibition. It also shows that Ser1021 falls outside the key interface occupied by Asp1019, but more could have been said about the potential effects of Ser1021Phe.Overall, I believe the authors fully achieved the aims of their study. The work will have a substantial impact because it reports the identification of novel disease-linked genes, and because it further demonstrates the high value of the Drosophila model for finding and understanding gene-disease linkages.
**Reviewer #3 (Public Review):**
Summary:The paper attempts to model the functional significance of variants of PLCG2 in a set of patients with variable clinical manifestations.Strengths:A study attempting to use the Drosophila system to test the function of variants reported from human patients.Weaknesses:Additional experiments are needed to shore up the claims in the paper. These are listed below.Major Comments:(1) Does the pLI/ missense constraint Z score prediction algorithm take into consideration whether the gene exhibits monoallelic or biallelic expression?

To our knowledge, pLI and missense Z don't consider monoallelic or biallelic expression. Instead, they reflect sequence constraint and are calculated based on the observed versus expected variant frequencies in population databases.

(2) Figure 1B: Include human PLCG2 in the alignment that displays the species-wide conserved variant residues.

We have updated Figure 1B and incorporated the alignment of PLCG2.

(3) Figure 4A:Given that(i) sl is predicted to be the fly ortholog for both mammalian PLCγ isozymes: PLCG1 and PLCG2 [Line 62](ii) they are shown to have non-redundant roles in mammals [Line 71](iii) reconstituting PLCG1 is highly toxic in flies, leading to increased lethality.This raises questions about whether sl mutant phenotypes are specifically caused by the absence of PLCG1 or PLCG2 functions in flies. Can hPLCG2 reconstitution in sl mutants be used as a negative control to rule out the possibility of the same?

The studies about the non-redundant roles of PLCG1 and PLCG2 mainly concern the immune system.

We have assessed the phenotypes in the *sl^T2A^/Y; UAS-hPLCG2* flies. Expression of human PLCG2 in flies is also toxic and leads to severely reduced eclosion rate.

We have updated the manuscript with these results, and included the eclosion rate of *sl^T2A^/Y; UAS-hPLCG2* flies in the new Figure 4B.

(4) Do slT2A/Y; UAS-PLCG1Reference flies survive when grown at 22{degree sign}C? Since transgenic fly expressing PLCG1 cDNA when driven under ubiquitous gal4s, Tubulin and Da, can result in viable progeny at 22{degree sign}C, the survival of slT2A/Y; UAS-PLCG1Reference should be possible.

The eclosion rate of *sl^T2A^/Y >PLCG1^Reference^* flies at 22°C is slightly higher than at 25°C, but remains severely reduced compared to the UAS-Empty control. We have presented these results in the updated Figure 4 - figure supplement 2.

and similarlyDoes slT2A flies exhibit the phenotypes of (i) reduced eclosion rate (ii) reduced wing size and ectopic wing veins and (iii) extra R7 photoreceptor in the fly eye at 22{degree sign}C?

The mutant phenotypes are still observed at 22 °C.

If so, will it be possible to get a complete rescue of the slT2A mutant phenotypes with the hPLCG1 cDNA at 22{degree sign}C? This dataset is essential to establish Drosophila as an ideal model to study the PLCG1 de novo variants.

Thank you for the suggestion. It is difficult to directly assess the rescue ability of the *PLCG1* cDNAs due to the toxicity. However, our ectopic expression assays show that the variants are more toxic than the reference with variable severities, suggesting that the variants are deleterious.

The ectopic expression strategy has been used to evaluate the consequence of genetic variants and has significantly contributed to the interpretation of their pathogenicity in many cases (reviewed in Her et al., Genome, 2024, PMID: 38412472).

(5) Localisation and western blot assays to check if the introduction of the de novo mutations can have an impact on the sub-cellular targeting of the protein or protein stability respectively.

Thank you for the suggestion.

We expressed *PLCG1* cDNAs in the larval salivary glands and performed antibody staining (rabbit anti-Human PLCG1; 1:100, Cell Signaling Technology, #5690). The larval salivary gland are composed of large columnar epithelia cells that are ideal for analyzing subcellular localization of proteins. The PLCG1 proteins are cytoplasmic and localize near the cell surface, with some enrichment in the plasma membrane region. The variant proteins are detected, and did not show significant difference in expression level or subcellular distribution compared to the reference. We did not include this data.

(6) Analysing the nature of the reported gain of function (experimental proof for the same is missing in the manuscript) variants:Instead of directly showing the effect of introducing the de novo variant transgenes in the Drosophila model especially when the full-length PLCG1 is not able to completely rescue the slT2A phenotype;(i) Show that the gain-of-function variants can have an impact on the protein function or signalling via one of the three signalling outputs in the mammalian cell culture system: (i) inositol-1,4,5-trisphosphate production, (ii) intracellular Ca2+ release or (iii) increased phosphorylation of extracellular signal-related kinase, p65, and p38.

We appreciate the reviewer’s suggestion. We utilized the CaLexA (calcium-dependent nuclear import of LexA) system (Masuyama et al., J Neurogenet, 2012, PMID: 22236090) to assess the intracellular Ca^2+^ change associated with the expression of PLCG1 cDNAs in fly wing discs. The results show that, compared to the reference, expression of the D1019G or D1165G variants leads to elevated intracellular Ca^2+^ levels, similar to the hyperactive S1021F and D1165H variants. However, the H380R or L597F variants did not show a detectable phenotype in this assay. These results suggest that D1019G and D1165G are hyperactive variants, whereas H380R and L597F variant are not, or their effect is too mild to be detected in this assay. We have updated the related sections in the manuscript and Figures 5A and Figure 5 - figure supplement 1.

OR(ii) Run a molecular simulation to demonstrate how the protein's auto-inhibited state can be disrupted and basal lipase activity increased by introducing D1019G and D1165G, which destabilise the association between the C2 and cSH2 domains. The H380R variant may also exhibit characteristics similar to the previously documented H335A mutation which leaves the protein catalytically inactive as the residue is important to coordinate the incoming water molecule required for PIP2 hydrolysis.

We utilized the DDMut platform, which predicts changes in the Gibbs Free Energy (ΔΔG) upon single and multiple point mutations (Zhou et al., Nucleic Acid Res, 2023, PMID: 37283042), to gain insight into the molecular dynamics changes of variants. The results are now presented in Figure S7.

Additionally, we performed Molecular dynamics (MD) simulations. The results show that, similar to the hyperactive D1165H variant, the D1019G and D1165G variants exhibit increased disorganization, with a higher root mean square deviations (RMSD) compared to the reference PLCG1. The data are also presented in the updated Figure 6 figure supplement 1.

(7) Clarify the reason for carrying out the wing-specific and eye-specific experiments using nub-gal4 and eyless-gal4 at 29°C despite the high gal4 toxicity at this temperature.

We used high temperature and high expression level to see if the mild H380R and L597F variants could show phenotypes in this condition.

The toxicity of the two strong variants (D1019G and D1165G) has been consistently confirmed in multiple assays at different temperatures.

(8) For the sake of completeness the authors should also report other variants identified in the genomes of these patients that could also contribute to the clinical features.

Thank you!

The additional variants and their potential contributions to the clinical features are listed and discussed in Table 1 and its legend.

**Reviewer #1 (Recommendations For The Authors):**
The manuscript's significant contribution is tempered by a lack of comprehensive analysis using the generated genetic reagents in Drosophila. To enhance our understanding of the PLCG1 orthologs, I suggest the following:(1) A more detailed molecular analysis to distinguish the actions of sl variants from the wild-type could be very informative. For example, utilizing the HA-epitope tag within the current UAS-transgenes could reveal more about the cellular dynamics and abundance of these variants, potentially elucidating mechanisms beyond gain-of-function.

We appreciate the reviewer’s suggestion. The *UAS-sl* cDNA constructs contain stop codon and do not express an HA-epitope tag. Alternatively, we utilized commercially available antibodies against human PLCG1 antibodies to assess the subcellular localization and protein stability by expressing the reference and variant PLCG1 cDNAs in Drosophila larval salivary glands. The reference proteins are cytoplasmic with some enrichment along the plasma membrane. However, we did not observe significant differences between the reference and variant proteins in this assay. We did not include this data.

(2) I suggest further investigating the relative contributions of developmental processes and acute (Adult) effects on the sl-variant phenotypes observed. For example, employing systems that allow for precise temporal control of gene expression, such as the temperature-sensitive Gal80, could differentiate between these effects, shedding light on the mechanisms that affect longevity and locomotion. This knowledge would be vital for a deeper understanding of the corresponding human disorders and for developing therapeutic interventions.

We appreciate the reviewer’s suggestion. We utilized *Tub-GAL4, Tub-GAL80ts* to drive the expression of *sl* wild-type or variant cDNAs, and performed temperature shifts after eclosion to induce expression of the cDNAs only in adult flies. The *sl^D1184G^* variant (corresponding to *PLCG1^D1165G^*) caused severely reduced lifespan and the flies mostly die within 10 days. The *sl^D1041G^* variant (corresponding to *PLCG1^D1019G^*) led to reduced longevity and locomotion. The *sl^H384R^* variant (corresponding to *PLCG1^H380R^*) showed only a mild effect on longevity and no significant effect on climbing ability. These results suggest that the two strong variants (*sl^D1041G^* and *sl^D1184G^*) contribute to both developmental and acute effects while the H384R variant mainly contributes to developmental stages.

I also suggest a more refined analysis of overexpression toxicity. Rather than solely focusing on ubiquitous transgene expression, overexpressing transgene in endogenous pattern using sl-t2a-Gal4 may yield a more nuanced understanding of the pathogenic mechanisms of gain-of-function mutations, particularly in the pathogenesis associated with these variants exclusively located in the coding regions.

We appreciate the reviewer’s suggestion. We therefore performed the experiments using *sl^T2A^* to drive overexpression of *PLCG1* cDNAs in heterozygous female progeny with one copy of wild-type *sl+ (sl^T2A^/ yw > UAS-cDNAs)*. In this context, expression of *PLCG1^Reference^*, *PLCG1^H380R^* or *PLCG1^L597F^* is viable whereas expression of *PLCG1^D1019G^* or *PLCG1^D1165G^* is lethal, suggesting that the *PLCG1^D1019G^* and *PLCG1^D1165G^* variants exert a strong dominant toxic effect while the *PLCG1^H380R^* and *PLCG1^L597F^* are comparatively milder. Similar patterns have been consistently observed in other ectopic expression assays with varying degrees of severity. These results are updated in the manuscript and figures.

**Reviewer #2 (Recommendations For The Authors):**
The work in the paper could be usefully extended by determining the effects of expressing His380Phe and His380Ala in flies. These variants suppress PLCG1 activity, so their phenotype, if any, would be predicted not to be the same as His380Arg. Determining this would add further strength to the conclusions of the paper.

We thank the reviewer for the constructive suggestions! We have tested the enzymatic-dead H380A variant, which still exhibits toxicity when expressed in *sl^T2A^/Y* hemizygous flies, but it is not toxic in heterozygous females suggesting that the reduced eclosion rate is likely not directly associated with enzymatic activity. We have updated the manuscript and figures accordingly.